# Cancer-associated hypersialylated MUC1 drives the differentiation of human monocytes into macrophages with a pathogenic phenotype

Richard Beatson [1✉], Rosalind Graham [1], Fabio Grundland Freile[1], Domenico Cozzetto [2], Shichina Kannambath[3], Ester Pfeifer[1], Natalie Woodman[4], Julie Owen[4], Rosamond Nuamah[3], Ulla Mandel [5], Sarah Pinder [6], Cheryl Gillett[4], Thomas Noll[7], Ihssane Bouybayoune[6], Joyce Taylor-Papadimitriou[1] & Joy M. Burchell [1✉]

The tumour microenvironment plays a crucial role in the growth and progression of cancer, and the presence of tumour-associated macrophages (TAMs) is associated with poor prognosis. Recent studies have demonstrated that TAMs display transcriptomic, phenotypic, functional and geographical diversity. Here we show that a sialylated tumour-associated glycoform of the mucin MUC1, MUC1-ST, through the engagement of Siglec-9 can specifically and independently induce the differentiation of monocytes into TAMs with a unique phenotype that to the best of our knowledge has not previously been described. These TAMs can recruit and prolong the lifespan of neutrophils, inhibit the function of T cells, degrade basement membrane allowing for invasion, are inefficient at phagocytosis, and can induce plasma clotting. This macrophage phenotype is enriched in the stroma at the edge of breast cancer nests and their presence is associated with poor prognosis in breast cancer patients.

[1] Breast Cancer Biology, Comprehensive Cancer Centre, King's College London, Guy's Cancer Centre, Guy's Hospital, London SE1 9RT, UK. [2] Translational Bioinformatics, Genomics Facility, National Institute for Health Research Biomedical Research Centre at Guy's and St Thomas' NHS Foundation Trust and King's College London, London SE1 9RT, UK. [3] Genomics Facility, National Institute for Health Research Biomedical Research Centre at Guy's and St Thomas' NHS Foundation Trust and King's College London, London SE1 9RT, UK. [4] KHP Tissue Bank, Breast Pathology, Comprehensive Cancer Centre, King's College London, Guy's Cancer Centre, Guy's Hospital, London SE1 9RT, UK. [5] Copenhagen Centre for Glycomics, Departments of Cellular and Molecular Medicine and Odontology, Faculty of Health Sciences, University of Copenhagen, Blegdamsvej 3, 2200N Copenhagen, Denmark. [6] Breast Pathology, Comprehensive Cancer Centre, King's College London, Guy's Cancer Centre, Guy's Hospital, London SE1 9RT, UK. [7] Cell Culture Technology, Faculty of Technology & CeBiTec, Bielefeld University, P.O. Box 10 01 31, 33501 Bielefeld, Germany. ✉email: richard.1.beatson@kcl.ac.uk; joy.burchell@kcl.ac.uk

The tumour metropolis consists of an ecosystem of tumour cells, stroma and infiltrating immune cells, and in breast cancers the tumour microenvironment (TME) can form 50% of the tumour mass. Tumour-associated macrophages (TAMs) make a considerable contribution to the TME and are associated with poor prognosis as demonstrated by a recent meta-analysis of sixteen studies in breast cancer[1]. TAMs contribute to all stages of cancer progression through a variety of mechanisms including promoting angiogenesis, inducing immune suppression and promoting inflammation[2,3]. Indeed, their importance in the initiation of mammary tumours has been shown by inducing premature recruitment of macrophages into the mammary gland which results in the promotion of malignancy[4], whereas depletion of macrophages can completely inhibit the growth of transplantable tumours[5].

In health, the majority of tissue resident macrophages are believed to originate from the erythroid-myeloid progenitors in the yolk sac, while most macrophages present in tumours are recruited from circulating monocytes[6]. Historically macrophages have been divided into M1-like which are pro-inflammatory and anti-tumour and M2-like which are involved in wound healing and thought to promote tumour growth. However, it is clear that this binary classification is no longer valid as data coming from RNAseq and single-cell RNAseq show transcriptional diversity and M1 and M2 defining genes expressed by the same cell[7–9]. Indeed, TAMs are phenotypically plastic, and factors produced by the cancer cells and the TME can induce macrophages to become tumour-promoting. These can include factors secreted by the tumour cells such as chemokines, cytokines and metabolites secreted and consumed within the TME[10].

Changes in glycosylation are common features of malignancy and often result in increased sialylation[11–14]. Members of the Siglec family of sialic acid binding lectins are expressed by many immune cells including monocytes and macrophages[15]. Siglecs are involved in regulation of the immune system and many contain immunoreceptor tyrosine-based inhibitory motifs. Indeed, recent studies have implicated binding of sialic acid to Siglecs as a means of cancer immune evasion[16,17,18].

MUC1 is a surface bound mucin that can be cleaved by proteases or shed, post-ligation, into the lumen. It is known to be over-expressed, de-polarised and aberrantly O-glycosylated in the majority of breast carcinomas. The alterations in O-glycosylation, from long branched chains to shorter structures, are primarily due to changes in glycosyltransferase expression[11–14]. These short glycans are frequently hypersialylated and we have shown that sialylation of the short trisaccharide (Neu5Acα2,3-Galβ1,3GalNAc) known as sialylated T (ST) leads to increased tumour growth in mouse models[19] and that this increased growth is immune cell dependent[20]. Moreover, MUC1 carrying the ST glycan is the dominant MUC1 glycoform found in sera of cancer patients[21]. Although the aberrant glycosylation resulting in MUC1 carrying the ST glycan has been known for many years and the conservation and high prevalence of this glycoform in breast and other adenocarcinomas suggested functionality, the mechanisms involved in its association with tumour progression have been poorly understood.

We and others have shown that MUC1 can bind to Siglec-9[22,23] that is expressed by monocytes, macrophages and some T cells[15,24]. We found that the sialylated tumour-associated glycoform of MUC1, MUC1-ST, bound to Siglec-9 expressed by monocytes and induced monocytes to secrete factors associated with tumour progression[22]. Here, we show that MUC1-ST is expressed by the majority of breast cancers and, acting in serum-free medium without the addition of cytokines, has the ability to induce the differentiation of monocytes to macrophages and to promote their viability. These macrophages show functional characteristics of TAMs, including potent basement membrane disruption, and can be identified in a specific region of primary breast cancers known to be associated with a worse prognosis. The transcriptional profile of these MUC1-ST-induced macrophages reveals a phenotype with multiple upregulated factors associated with poor prognosis, and defines a signature associated with poor survival of breast cancer patients.

## Results

**The ST glycoform of MUC1 is very common in breast cancers and correlates with stromal macrophage infiltrate.** Analysis of 53 whole primary breast cancers showed that a sialylated glycoform of MUC1, MUC1-ST (which carries the glycan, Neu5Acα2, 3Galβ1, 3GalNAc), was expressed by 83% of breast cancers (Fig. 1a, b). Analysis of the breast cancer subtypes showed that triple negative breast cancers had the lowest expression of MUC1-ST and oestrogen receptor-positive breast cancers the highest (Supplementary Fig. 1a). Given the high expression of MUC1-ST in breast cancers, the well-established impact of macrophage presence, and that MUC1-ST can bind to Siglec-9 expressed by macrophages[22], we analysed cases for macrophage infiltrate and assessed for any association with MUC1-ST.

Initially we documented the location of CD163+ macrophages, finding a higher number of macrophages on the edge of the tumour nests (Fig. 1c, Supplementary Fig. 1b). Figure 1d shows examples of two cases with high and low expression of MUC1-ST and the staining of consecutive sections by IHC of CD163. Scoring of macrophages in different geographical regions by manual (Fig. 1e) and automated (using Visiopharm software; Supplementary Fig. 1c) methodologies revealed a significant association between MUC1-ST and CD163 on the edge of the tumour nests. As there was no correlation between MUC1-ST and tumour-derived cerebrospinal fluid 1 (CSF1) (Supplementary Fig. 1d), we hypothesised that MUC1-ST itself may be able to drive macrophage differentiation in this specific location.

**MUC1-ST alone can induce primary healthy monocytes to differentiate into macrophages with a TAM-like phenotype.** Given the findings in Fig. 1 and the fact that MUC1-ST can bind to and activate monocytes[22], we assessed whether MUC1-ST alone could drive the differentiation of monocytes into macrophages. Monocytes isolated from the peripheral blood mononuclear cells (PBMCs) of healthy donors were treated with MUC1-ST, MUC1-ST treated with sialidase to remove the sialic acid (MUC1-T) or M-CSF as a control, all in serum-free medium for 7 days. Figure 2a, b shows that MUC1-ST supported the viability of macrophages similar to M-CSF but this was not observed when the sialic acid was removed from the MUC1-ST (MUC1-T). This indicates that MUC1-ST was binding to Siglecs, and indeed the binding to monocytes could be inhibited by over 90% in the presence of anti-Siglec-9 (Supplementary Fig. 2a) as previously reported[22]. Less than 20% inhibition with anti-Siglec-7 was observed at the maximum concentration of antibody (Supplementary Fig. 2a). Phenotypic analysis showed that MUC1-ST treated monocytes expressed TAM-like markers, showing significantly higher levels of PD-L1 and CD206 than M-CSF treated monocytes or monocytes treated with MUC1-T and so lacking sialic acid (Fig. 2c). MUC1-ST-treated monocytes also showed expression of CD163 and low levels of CD86 (Fig. 2c). Moreover, the induction of this phenotype by MUC1-ST was dose dependent (Supplementary Fig. 2b).

Given that treatment of monocytes with MUC1-ST can induce the secretion of M-CSF (Supplementary Fig. 2c), monocytes were cultured with M-CSF or MUC1-ST in the presence of an M-CSF neutralising antibody or isotype control. While there was a total

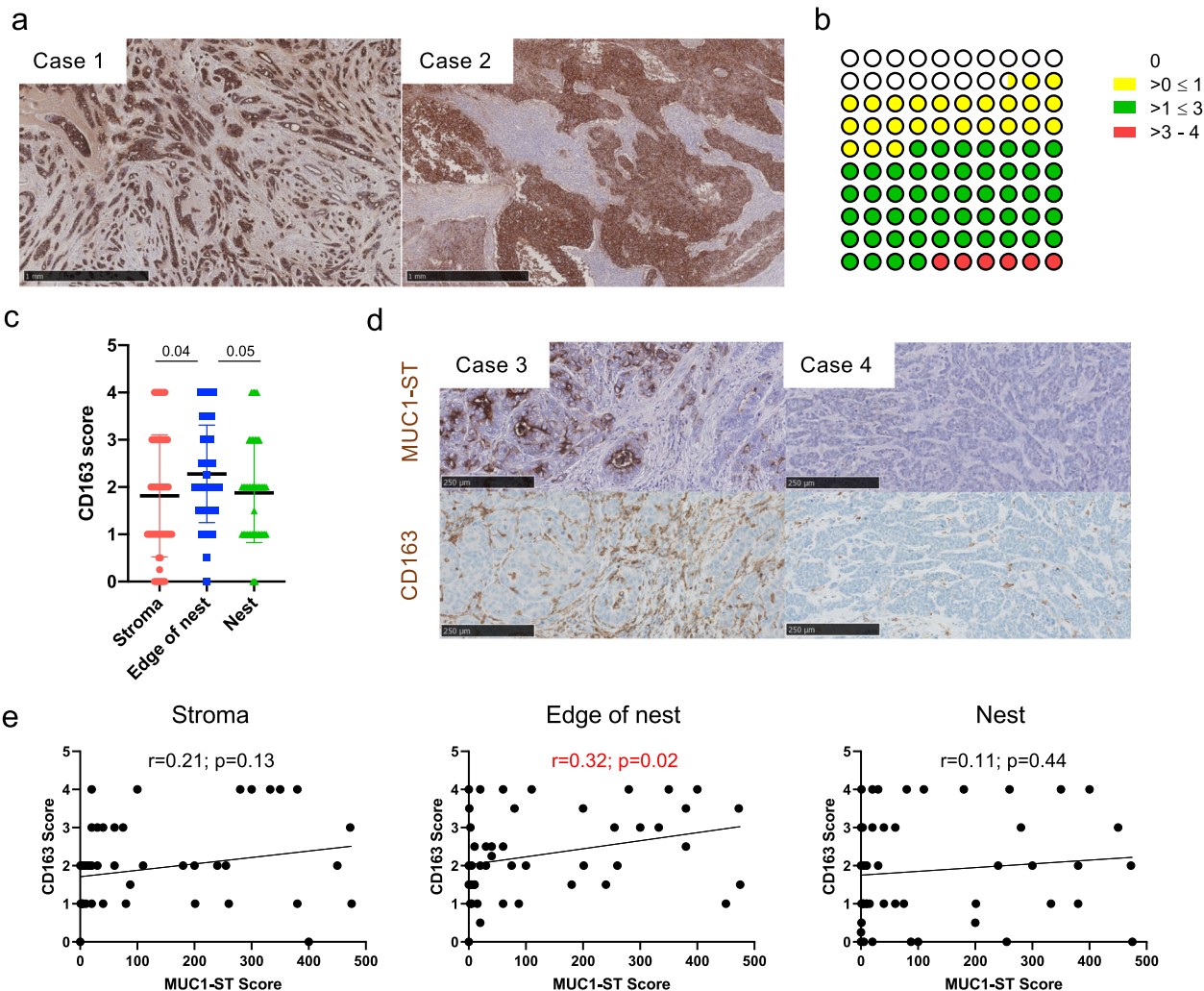

**Fig. 1 MUC1-ST is expressed by most breast cancers and its expression positively correlates with macrophage presence on the edge of tumour nests. a** Examples of positive MUC1-ST IHC staining in breast cancers (a negative example is included in (**d**)). **b** Summary of manual tissue scoring of MUC1-ST expression in breast cancers ($n = 53$ breast cancer cases). **c** CD163 manual scores in different regions of breast cancers ($n = 53$ breast cancer cases). **d** Examples of sequential sections stained for MUC1-ST (brown) and CD163 (brown) by IHC. **e** CD163 scores in different indicated regions of the tumour measured against MUC1-ST scoring ($n = 53$ breast cancer cases). Standard error of the mean shown and paired $t$ test used for statistical analysis. Correlations were analysed using linear regression analysis (Pearson's).

lack of viable cells when monocytes were cultured with M-CSF in the presence of the M-CSF neutralising antibody, this antibody had no effect on the viability or number of MUC1-ST cultured cells, nor on their phenotype (Fig. 2d–f). Thus, factors other than M-CSF were supporting the differentiation of the MUC1-ST-induced macrophages. We have previously shown that MUC1-ST binding to monocytes did not induce phosphorylation of Siglec-9 or SHP-1, which is associated with inhibitory signalling. In contrast down-stream activation of the MEK-ERK pathway occurred[22]. We therefore treated monocytes with a MEK/ERK inhibitor (PD98059) prior to the addition of MUC1-ST and found that differentiation was profoundly inhibited (Supplementary Fig. 2d).

**The transcriptome of MUC1-ST-induced macrophages is different to M-CSF-induced macrophages**. As MUC1-ST supported the differentiation of monocytes to TAM-like macrophages, this glycoform is commonly expressed in breast cancers and correlated with macrophages present in the stroma around the cancer nests, we wished to further explore the

relationship between MUC1-ST and TAMS. RNAseq was performed on MUC1-ST-induced macrophages and compared to donor matched M-CSF-induced macrophages. Monocytes from three healthy donors were treated with M-CSF or MUC1-ST for 7 days in serum-free medium, viable cells sorted, the RNA isolated, and RNAseq performed. The expressed genes are documented in Supplementary Data 1 and deposited in GEO reference GSE150613. Application of CIBERSORT[25] analysis to the starting monocytes and the MUC1-ST or M-CSF induced macrophages confirmed the monocyte-derived macrophage immune subtype of the MUC1-ST-induced cells as M0-like (Supplementary Fig. 3a). Figure 3a, b shows the hierarchical clustering and t-sne plots of the samples, and Fig. 3c the volcano plots of the transcripts after differential analysis comparing matched MUC1-ST-induced macrophages and M-CSF macrophages. These data illustrate that M-CSF and MUC1-ST-induced macrophages express a very different profile of genes. Also shown are the top and bottom 50 genes differentially expressed by MUC1-ST-induced macrophages (Fig. 3d, e). *CXCL5* was one of the top differentially expressed genes in the MUC1-ST-induced macrophages, and *SERPINE1*/PAI-1 was a high differential (Fig. 3f). PAI-1 has been

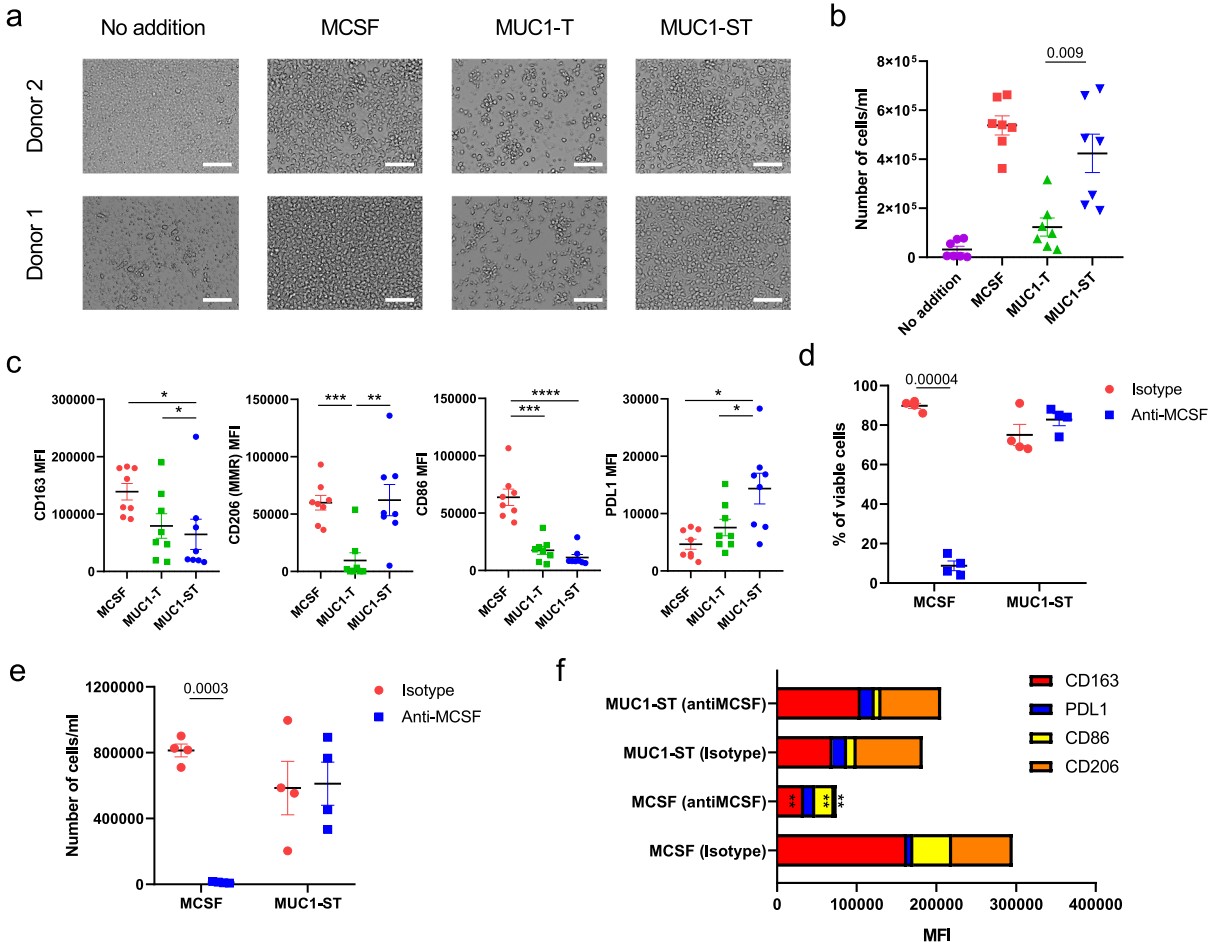

**Fig. 2 Recombinant MUC1-ST can induce the differentiation of monocytes into TAMs in an M-CSF independent manner. a** Bright field images of representative healthy donor monocytes treated with indicated factors for 7 days in serum-free media. **b** Number of viable cells from monocytes treated with indicated factors for 7 days in serum-free media ($n = 7$ biologically independent samples). **c** Phenotype of monocytes treated with indicated factors for 7 days in serum-free media ($n = 8$ biologically independent samples). **d**, **e** % viable cells (**d**), and number of viable cells (**e**) from monocytes cultured with either MCSF or MUC1-ST for 7 days in serum-free media in the presence of anti-MCSF neutralising antibodies or isotype control ($n = 4$ biologically independent samples). **f** Summary of phenotype for cells in (**d**). Standard error of mean shown and paired $t$ test used for statistical analysis. *$p < 0.05$, **$p < 0.01$, ***$p < 0.001$, ****$p < 0.0001$.

associated with carcinogenesis and was one of the factors we previously showed to be induced when MUC1-ST binds to monocytes[22]. Moreover, as both are secreted factors, like many of the top differentials, we reasoned that secreted factors may have the greatest local influence and therefore validated the expression of these mRNAs at the protein level as shown in Fig. 3g. Importantly, the expression of CXCL5 by MUC1-ST-induced macrophages was significantly reduced when MUC1-ST was stripped of its sialic acid (Fig. 3h) and the expression was also significantly inhibited by a Siglec-9 antibody (Fig. 3i). The expression of PAI-1 also showed similar trends. Moreover, when monocytes were co-cultured with the breast cancer line T47D that carries the MUC1-ST glycoform[14,26], CXCL5 was secreted by the myeloid cells and was reduced when the T47D cells were treated with sialidase to remove the sialic acid (Fig. 3j). Furthermore, monocytes cultured in the presence of T47D cells that had been engineered so that MUC1 carries long, branched chains rather than ST[26] showed a reduction in the secretion of CXCL5 (Fig. 3k). Further evidence for the requirement of sialic acid on MUC1-ST is shown in Supplementary Fig. 3 where a further three validated genes (Supplementary Fig. 3b, c) showed reduced expression when sialic acid is removed from MUC1-ST

(Supplementary Fig. 3d). Furthermore, the addition of a Siglec-9 antibody during the differentiation also reduces the expression of these three proteins (Supplementary Fig. 3e).

CXCL5 and CD206 (MMR) were inhibited by the use of a MEK/ERK inhibitor prior to initial stimulation with MUC1-ST (Supplementary Fig. 3f) and this is likely to be due to the impact on differentiation observed in Supplementary Fig. 2d. However, it does further highlight the dependency of these processes on these kinases. Supplementary Fig. 3g shows that the expression of a further 17 genes and 15 of these were validated at the protein level. Importantly, PD-L1 was highly significantly upregulated in MUC1-ST-induced macrophages. Intriguingly, when assessing the difference in Siglec transcript expression, most Siglecs were downregulated, including Siglec-9, which did not however reach significance ($p = 0.077$, Fig. 3c). The only transcripts showing profound significance were Siglecs 1, 14 and 16 which were all downregulated (Supplementary Fig. 3h). Siglec 1 has no intracellular signalling motif, whilst Siglecs 14 and 16 are both activating Siglecs[24]. However, the blocking experiments (Supplementary Fig. 2a) showed that MUC1-ST binding to Siglec-9 plays a dominant, but perhaps not exclusive role, in the profile of gene expression observed in MUC1-ST-induced macrophages.

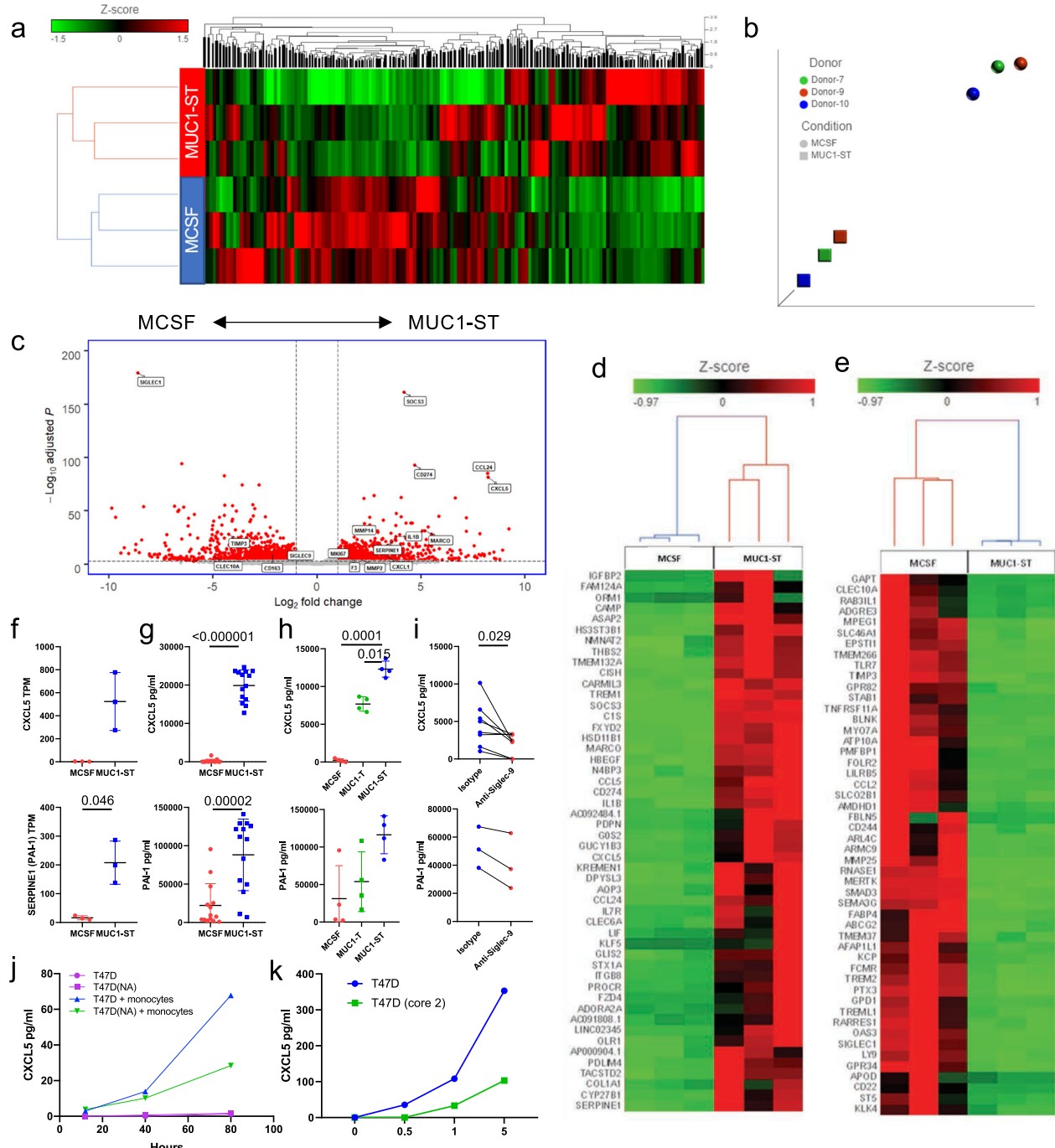

**Fig. 3 MUC1-ST induced a particular TAM phenotype. a** Hierarchical clustering of all transcripts from matched MUC1-ST ($n = 3$ biologically independent samples) and MCSF ($n = 3$ biologically independent samples) treated monocytes after RNAseq and Partek flow analysis. **b** t-sne plot showing clustering of MUC1-ST and MCSF treated monocyte transcriptomes ($n = 3$). **c** Volcano plot showing the fold change and significance (FDR) of differentially expressed genes in matched MCSF and MUC1-ST macrophages ($n = 3$ biologically independent samples). Low expressed genes removed (see Supplementary Data 1; tab 3). **d** Top 50 differentially expressed genes between matched MUC1-ST and MCSF macrophages. Low expressed genes removed (see Supplementary Data 1; tab 3). **e** As (**d**) but bottom 50 expressed genes. **f** CXCL5 and *SERPINE1* transcript expression in matched monocytes treated with MCSF ($n = 3$ biologically independent samples) or MUC1-ST ($n = 3$ biologically independent samples) for 7 days in serum-free media. **g** CXCL5 and PAI-1 protein levels in the supernatant of monocytes treated with MUC1-ST or MCSF for 7 days in serum-free media ($n = 14$ biologically independent samples). **h** CXCL5 and PAI-1 levels in the supernatant of monocytes treated with MUC1-ST or desialylated MUC1-ST (MUC1-T) for 7 days in serum-free media ($n = 4$ biologically independent samples). **i** CXCL5 ($n = 8$ biologically independent samples) and PAI-1 ($n = 3$ biologically independent samples) levels in the supernatant of MUC1-ST macrophages pre-treated with anti-Siglec-9 antibodies or isotype control. **j** CXCL5 levels in the supernatant of monocyte/T47D (±neuraminidase pre-treatment; NA) cocultures after 48 h of co-culture at a 5:1 ratio, $n = 2$ biologically independent samples with technical triplicate. **k** CXCL5 levels in the supernatant of monocyte/T47D or monocyte/T47D (core 2) cocultures after 48 h of co-culture at a 5:1 ratio. Representative of two independent experiments. Standard error of mean shown and paired *t* test used for statistical analysis. TPM transcripts per million.

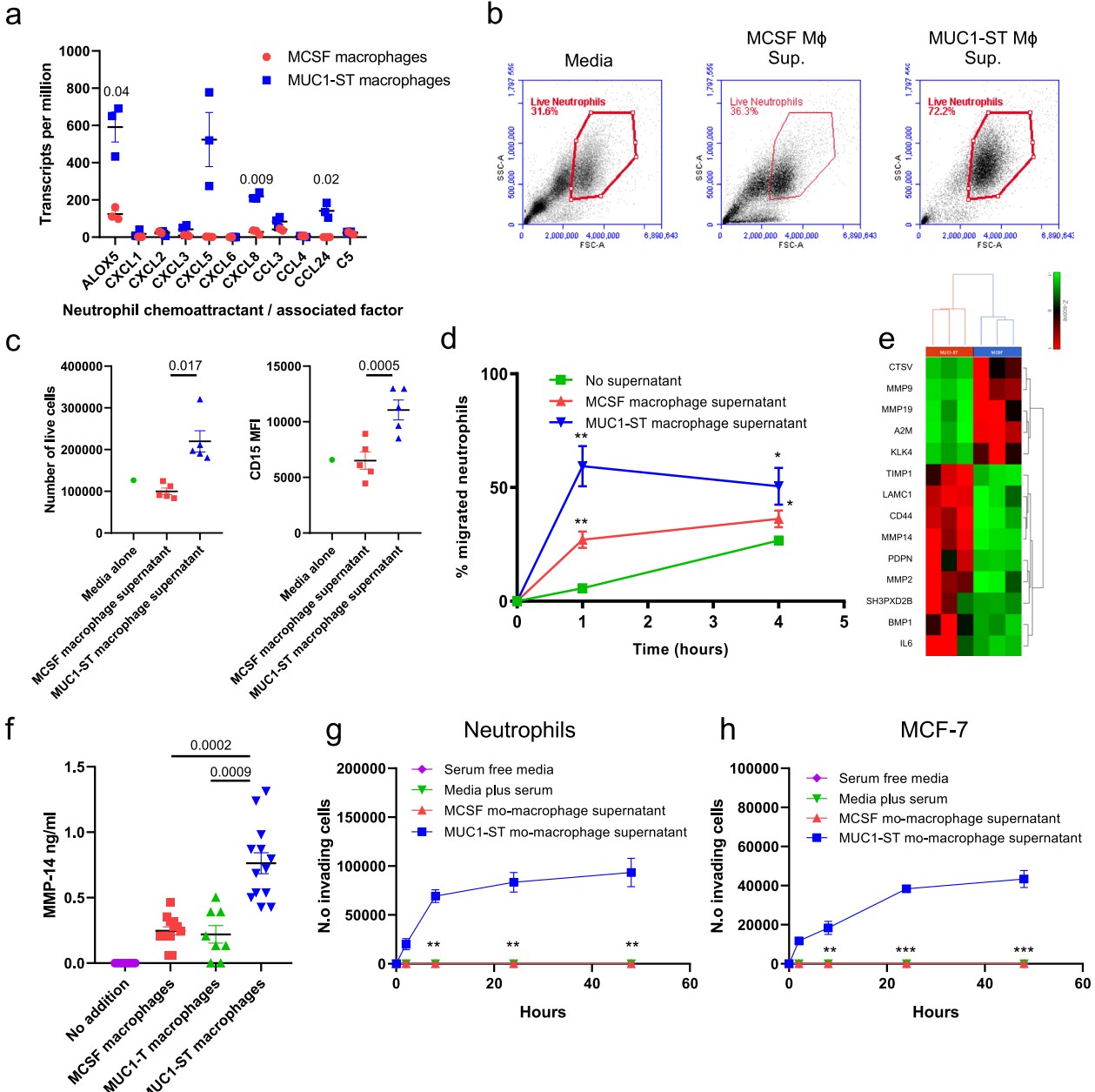

**Fig. 4 MUC1-ST macrophages can sustain neutrophils and induce their migration and invasion. a** Neutrophil chemoattractant or associated factor transcript expression level in MUC1-ST macrophages ($n = 3$ biologically independent samples) and MCSF macrophages ($n = 3$ biologically independent samples). **b** Example FSC/SSC plots of primary neutrophils 48 h after being cultured in indicated media or supernatant ($n = 5$ biologically independent samples). 'Live neutrophils' were defined as live using a viability dye. **c** Numbers and phenotype of live neutrophils 48 h after being cultured in indicated media or supernatant ($n = 5$ biologically independent samples). 'Live cells' were defined as live using a viability dye. **d** Migration of neutrophils towards indicated media or supernatant over indicated time period ($n = 5$ biologically independent samples). **e** Heatmap showing differentially expressed extracellular matrix disassembly genes (GO:0022617) in MUC1-ST ($n = 3$ biologically independent samples) and MCSF ($n = 3$ biologically independent samples) macrophages. **f** MMP14 protein levels in supernatant of monocytes treated with indicated factors for 7 days ($n = 13$ biologically independent samples; desialylated MUC1-ST, MUC1-T, $n = 8$ biologically independent samples). **g** Number of neutrophils invading through basement membrane extract towards the indicated media or supernatant at the indicated time points ($n = 5$ biologically independent samples). **h** Number of breast cancer cells (MCF-7) invading through basement membrane extract towards the indicated media or supernatant at the indicated time points ($n = 5$ biologically independent samples). Standard error of mean shown and paired $t$ test used for statistical analysis.

## MUC1-ST-induced macrophages have distinct functional capabilities

*Neutrophil function.* Neutrophils have been shown to contribute both to breast cancer metastasis[27–30] and to anti-tumour responses[31–33]. A number of chemokines such as CXCL5, CXCL8 and CCL24[28,30] that are differentially expressed in MUC1-ST-induced macrophages compared to M-CSF macrophages are involved in neutrophil recruitment (Fig. [4]a). Leukotrienes also have a chemotactic effect on neutrophils[34] and ALOX5 which catalyses the first step in leukotriene synthesis is also upregulated in MUC1-ST-induced macrophages compared to M-CSF-induced macrophages (Fig. [4]a). Therefore, neutrophils

isolated from healthy donors were cultured in the supernatant from MUC1-ST-induced macrophages or M-CSF macrophages. MUC1-ST macrophage supernatant was able to maintain the viability of 72% of the neutrophils at 48 h in comparison to M-CSF macrophage supernatant that was no better than medium alone (Fig. 4b, c). Moreover, the expression of CD15, which is associated with neutrophil maturation[35], was elevated on neutrophils incubated with supernatant from MUC1-ST-induced macrophages (Fig. 4c). Supernatant from MUC1-ST-induced macrophages also significantly increased the migration of neutrophils compared to M-CSF macrophage supernatant (Fig. 4d).

*Invasion.* MUC1-ST-induced macrophages expressed genes associated with extracellular matrix disassembly, particularly MMP14, the expression of which is dependent on the sialic acid carried on MUC1-ST (Fig. 4e, f). As macrophages mediate basement membrane degradation to promote invasion and metastasis[36,37], the invasion of neutrophils and cancer cells through basement membrane extract towards the various supernatants was investigated. Figure 4g shows that while control media and M-CSF-induced macrophage medium induced no invasion of neutrophils, supernatant from MUC1-ST-induced macrophages induced a significant number of cells to invade through basement membrane within 2 h. Moreover, supernatant from MUC1-ST-induced macrophages induced the invasion through the basement membrane of the breast cancer cell line, MCF-7, in a similar manner (Fig. 4h).

*Clotting.* Cancer patients are at a higher risk of developing serious bloods clots and breast cancer patients are at a risk of developing venous thromboembolism[38]. Two genes associated with blood coagulation, coding for factor 8 (*F8*) and tissue factor (*F3*) were also found to be differentially expressed by MUC1-ST-induced macrophages (Fig. 5a). Therefore, the expression of tissue factor by MUC1-ST- and M-CSF-induced macrophages was investigated. While there was no difference in the surface expression of tissue factor between MUC1-ST- and M-CSF-induced macrophages (Fig. 5b), MUC1-ST-induced macrophages secreted significantly more tissue factor than M-CSF macrophages and there was a requirement for sialic acid (Fig. 5c). Moreover, supernatant from MUC1-ST-induced macrophages induced significantly faster clotting than M-CSF macrophage supernatant (Fig. 5d).

*Phagocytosis.* A number of genes associated with phagocytosis (e.g. CD36) were significantly downregulated in MUC1-ST-induced macrophages although the expression of some scavenger receptor genes such as MARCO which has been associated with a poor prognosis in breast cancer[39] were significantly upregulated (Supplementary Fig. 3f, Supplementary Data 1). The phagocytic ability of the MUC1-ST-induced macrophages was therefore investigated. Figures 5e, f shows that MUC1-ST-induced macrophages were significantly less efficient at phagocytosis compared to M-CSF macrophages of both dextran (Fig. 5e) and a breast cancer cell line compared to M-CSF macrophages (Fig. 5f, Supplementary Fig. 4a).

*T-cell function.* Genes associated with the inhibition of T-cell function such as PD-L1 and IDO and protein expression of arginase were upregulated in MUC1-ST-induced macrophages whereas CD86 was downregulated (Supplementary Data 1, Supplementary Figs. 3 and 4b). We therefore investigated the ability of MUC1-ST-induced macrophages to inhibit T-cell proliferation. Indeed, supernatant from MUC1-ST monocytes significantly reduced the proliferation and viability of anti-CD3 stimulated PBMC and the proliferation and viability of PBMC in a mixed lymphocyte reaction (Fig. 5g, h, Supplementary Fig. 4c).

Taken together these data indicate that macrophages induced by MUC1-ST show functional characteristics of TAMs, in that they recruit and prolong the lifespan of neutrophils, degrade basement membrane, are inefficient at phagocytosis and inhibit T-cell proliferation and viability. Moreover, these macrophages can promote blood clotting.

**MUC1-ST-induced macrophages are present in primary breast cancer and associated with poor prognosis.** To investigate the presence of MUC1-ST-induced macrophages in primary breast cancer, the expression of *SERPINE1* (PAI-1) which is differentially expressed by MUC1-ST-induced macrophages (Fig. 3f, g) was measured by RNAscope on consecutive sections. Figure 6a, b shows that *SERPINE1* is upregulated in breast cancer and that significantly higher expression is found in the stroma around the edges of the nests of cancer cells compared to within the cancer cell nests or the stroma around the tumour (Fig. 6b). Moreover, *SERPINE1* expression in cells found in the stroma around the edges of the cancer cell nests is significantly correlated with MUC1-ST expression (Fig. 6c).

In addition, 24 primary breast cancers were double stained for CD68 and CXCL5 (Fig. 6d). Importantly, CD68 macrophages expressing CXCL5 were found within the cancers and with significantly higher numbers in the stroma around the nests of cancer cells (Fig. S5a). Moreover, there was a trend that CD68[+]CXCL5[+] macrophages in the stroma around the edges of the cancer nests to be associated with MUC1-ST expression (Fig. 6e).

Analysis of the TCGA breast cancer database shows a highly significant correlation between CD163 or CD68 and SIGLEC9 but not with the epithelial markers, EPCAM or KRT8 (Fig. S5b). Moreover, BASEscope analysis of our cohort of breast cancer showed expression of SIGLEC9 within the stroma, edge and nest of the tumour (Fig. S5c) in a similar manner to CD163 staining. Encouragingly, SIGLEC9 expression showed a trend for an inverse correlation with MUC1-ST expression suggesting the down regulation of the receptor upon engagement (Supplementary Fig. 5d), which was also observed at both the RNA and protein level in our in vitro studies (Supplementary Fig. 3f).

Finally, to determine whether these proteins may be present in the TME, we assessed for seven top validated factors, and MUC1, in the interstitial fluid of fresh breast cancers (Supplementary Fig. 5e), finding all factors, to varying levels, in all tumours tested.

As MUC1-ST-induced macrophages were able to recruit and prolong the lifespan of neutrophils, inhibit T-cell responses and enable cellular invasion through basement membrane extract, we investigated if MUC1-ST expression or MUC1-ST macrophage presence were associated with poor prognosis in breast cancers. Firstly, determining the expression of the top ten prognostic genes associated with a poor or favourable prognosis in all cancers identified by Gentles et al.[25], we showed that 8 out of 10 genes associated with poor prognosis were upregulated by MUC1-ST-induced macrophages compared to M-CSF macrophages (Fig. 7a). In contrast, four of the genes associated with a good prognosis were differentially upregulated by M-CSF-induced macrophages (Fig. 7b). Secondly, we had data on lymph node involvement for 20 patients in our cohort, and we observed a significant correlation between the percentage of involved lymph nodes and the expression of MUC1-ST (Fig. 7c, d). Finally, we assessed whether a MUC1-ST macrophage gene signature consisting of the top nine differentially expressed genes was associated with clinical outcome using the TCGA database. Figure 7e, f shows a highly significant correlation between a high MUC1-ST macrophage signature and shorter disease-free and overall survival.

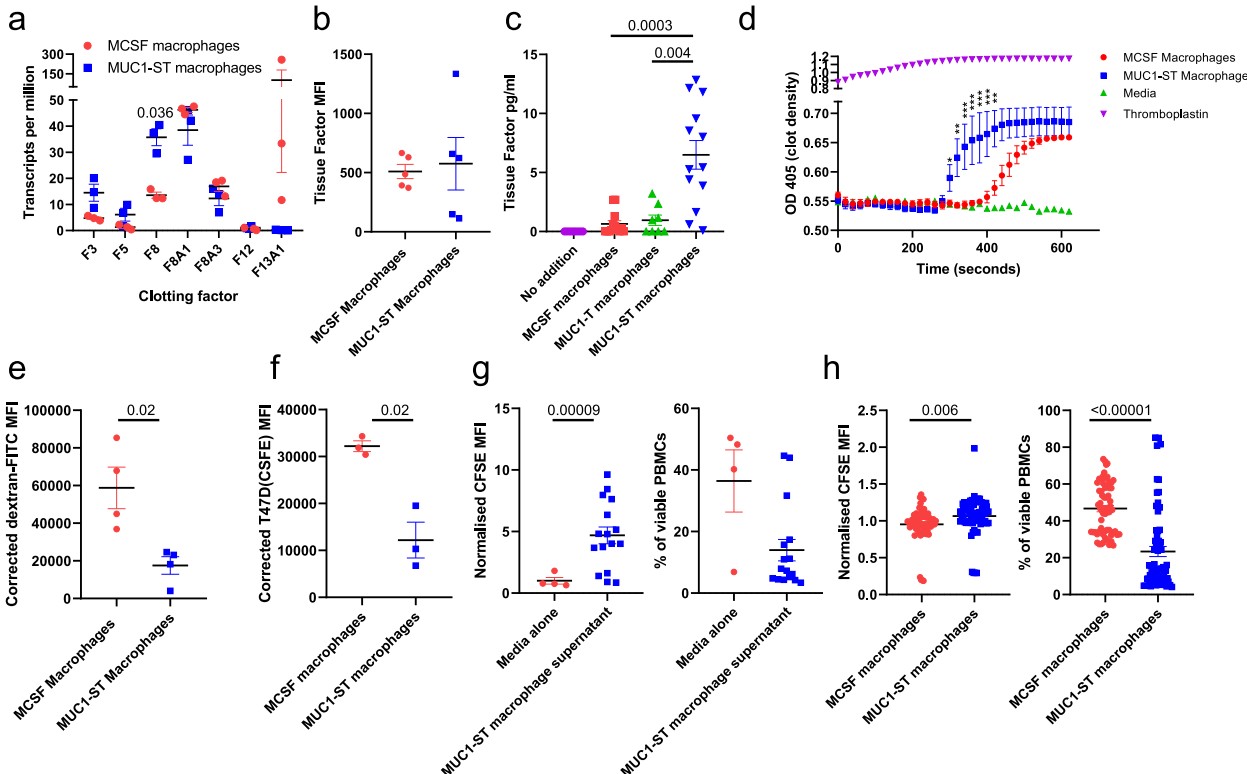

**Fig. 5 MUC1-ST macrophages induce clotting, are inefficient at phagocytosis and inhibit T-cell proliferation and viability. a** Clotting factor transcript expression levels in MUC1-ST ($n = 3$ biologically independent samples) and MCSF macrophages ($n = 3$ biologically independent samples). **b** Cell surface bound ($n = 5$ biologically independent samples) and **c** secreted levels of tissue factor ($n = 13$ biologically independent samples); excluding MUC1-T where $n = 8$ biologically independent samples. **d** Plasma clotting in the presence of indicated factors or supernatants at indicated time points. **e, f** Bar charts showing corrected (37 °C MFI minus 4 °C MFI) of **e** dextran-FITC uptake ($n = 4$ biologically independent samples) and **f** uptake of CFSE-labelled T47D tumour cells ($n = 3$ biologically independent samples) by M-CSF macrophages and MUC1-ST macrophages after 4 h incubation. **g** Pooled data showing proliferation (relative CFSE expression) and viability of CD3 stimulated PBMCs in the presence of media alone ($n = 4$ biologically independent samples) or MUC1-ST macrophage supernatant ($n = 16$ biologically independent samples). **h** Mixed leucocyte reaction showing proliferation and viability of PBMCs when co-cultured with MCSF macrophages or MUC1-ST macrophages at a 5:1 ratio for 4 days ($n = 16$ biologically independent samples, in quadruplet). Standard error of the mean shown and paired $t$ test used for statistical analysis, $**p < 0.01$, $***p < 0.001$.

## Discussion

Aberrant glycosylation, often resulting in hypersialylation is a common feature of cancer[11–14] and this has been shown to lead to the engagement of Siglecs[16–18,22,23]. The MUC1 mucin which carries multiple O-linked glycans shows a dramatic change in glycosylation in many cancers, including breast cancers, resulting in the core protein carrying multiple sialylated tri-saccharides known as ST. Here. we have shown that MUC1-ST in serum-free medium, and in the absence any other factor, can induce monocytes to differentiate into macrophages with a unique phenotype that to the best of our knowledge has not previously been described. The requirement for sialic acid on MUC1 and the data using a Siglec-9 antibody to block the interaction, indicate that MUC1-ST-induced macrophages are induced through the engagement of Siglec-9 expressed by monocytes. Previous data have shown that when MUC1-ST binds to Siglec-9, phosphorylation of Siglec-9 is reduced, evoking calcium flux and activation of the MEK-ERK pathway[22]. Here, we find that the ability of MUC1-ST to drive macrophage differentiation is MEK-ERK dependent. Further work is required to elucidate exactly how the engagement of what is considered an inhibitory Siglec, promotes such $Ca^{2+}$ and MEK-ERK dependent responses. However, Siglec-9 and other CD33-like Siglecs do contain a well-conserved activating SLAM-like domain with no known function[40]. Moreover, several studies have shown the *cis*-binding of Siglecs to activating receptors, such as TLR4, results in the formation of complexes

that alters activation[41–43]. It is possible that MUC1 binding could break such complexes resulting in activation of a receptor[44]. The transcripts of activating Siglecs 14 and 16 are significantly decreased in MUC1-ST-induced macrophages, therefore we cannot exclude the possibility that these Siglecs may also have a role in driving these observations. However, given the data that over 90% of the binding of MUC1-ST to monocytes can be inhibited by blocking Siglec-9 (Supplementary Fig. 2a) this seems unlikely. Finally, a recent publication investigating another hypersialylated structure, glycodelin-A, in pregnancy, found it was able to drive similar macrophage phenotypes as we have previously observed[22], although through Siglec-7, not Siglec-9[45].

We applied CIBERSORT[25] to the transcriptome of these MUC1-ST-induced macrophages and confirmed their macrophage phenotype, we then validated 22/24 differential hits. These macrophages generated in vitro showed the functional characteristics of TAMs in that they are inefficient at phagocytosis, inhibit T-cell proliferation, recruit neutrophils and promote invasion. Analysis of 53 breast cancers demonstrated the presence of this macrophage subtype in primary breast cancers and using the top nine differentially expressed genes by the MUC1-ST-induced macrophages, we showed a significant association with poor prognosis. Interestingly, a recent paper has shown a significant correlation between 'cancer-associated MUC1' (a mixture of glycophenotypes) and macrophages when staining with

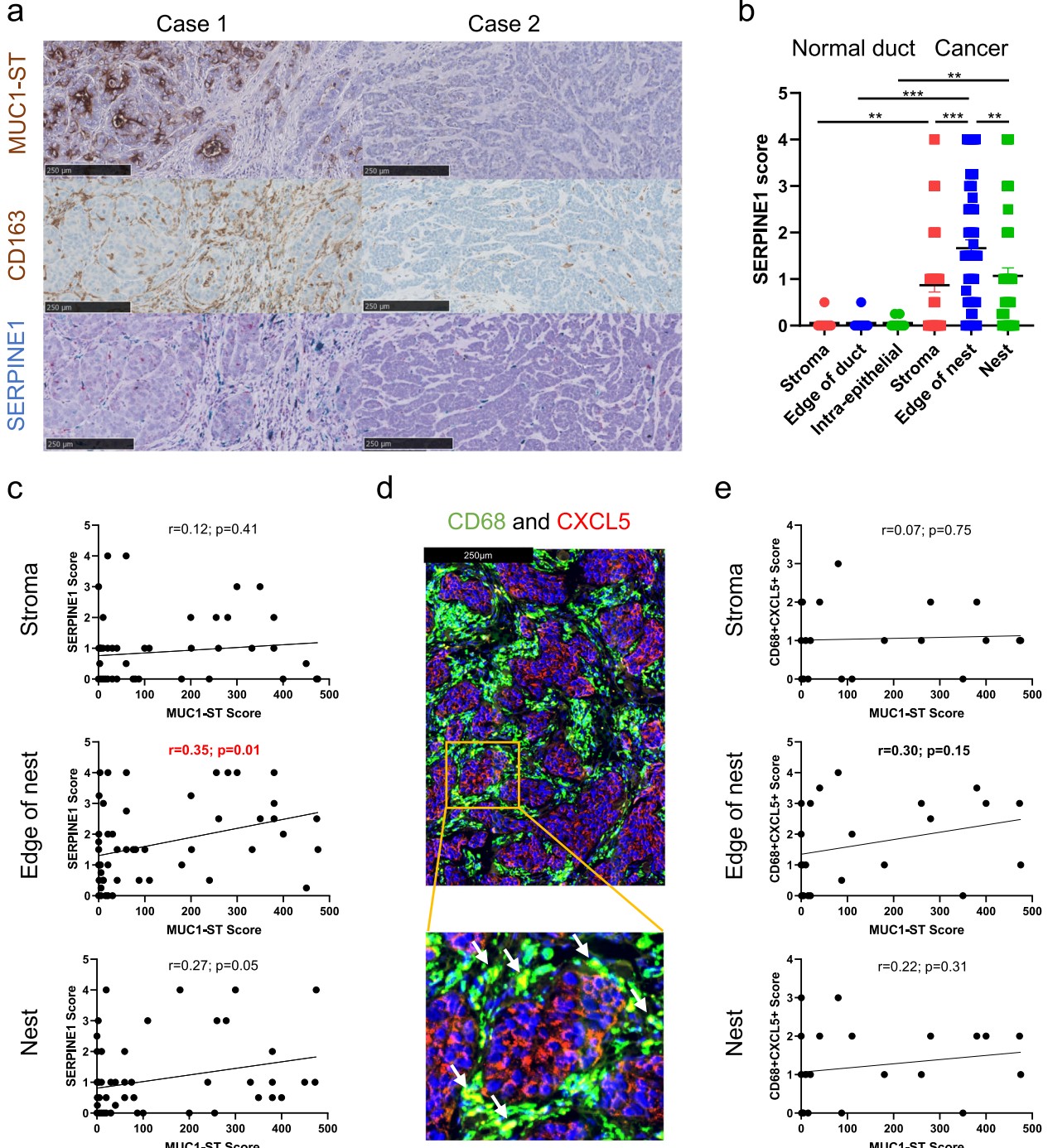

**Fig. 6 MUC1-ST macrophages are present in the stroma surrounding primary breast cancers. a** Sequential sections stained for MUC1-ST (brown) and CD163 (brown) by IHC and *SERPINE1* (blue/green) by RNAscope. **b** Manual scoring of *SERPINE1 expression* in different indicated regions of healthy ducts (within $n = 12$ of breast cancer cases) and tumours ($n = 53$ breast cancer cases). **c** *SERPINE1* manual scores in different indicated regions of the tumour measured against MUC1-ST manual scoring ($n = 53$ breast cancer cases). **d** Example image of CD68+ CXCL5+ double staining; double positive cells are displayed as yellow as indicated by arrows. **e** CD68+ CXCL5+ manual scores in different indicated regions of the tumour measured against MUC1-ST scoring ($n = 24$ breast cancer cases). Standard error of the mean shown and paired *t* test used for statistical analysis. Correlations were analysed using linear regression analysis (Pearson's). $**p < 0.01$, $***p < 0.001$.

CD163[46]. Our data indicate that at least one of the mechanisms whereby MUC1-ST can affect progression of cancers is by the direct induction of monocyte differentiation into macrophages with a TAM-like phenotype without the need for any additional factor.

The presence of TAMs being pro-tumoral is now well established in breast cancer, and a meta-analysis of 16 studies

demonstrated that high density of TAMs is associated with a poor prognosis[1]. Moreover, the specific location of TAMs within a tumour is known to have an impact on their pro-tumour activity. It is the TAMs outside the nests in the stroma rather than within the nest of the cancer cells that are associated with the worst outcome. Indeed, CD163 or CD68 macrophages in the stroma rather than in the cancer cell nests have been shown to correlate

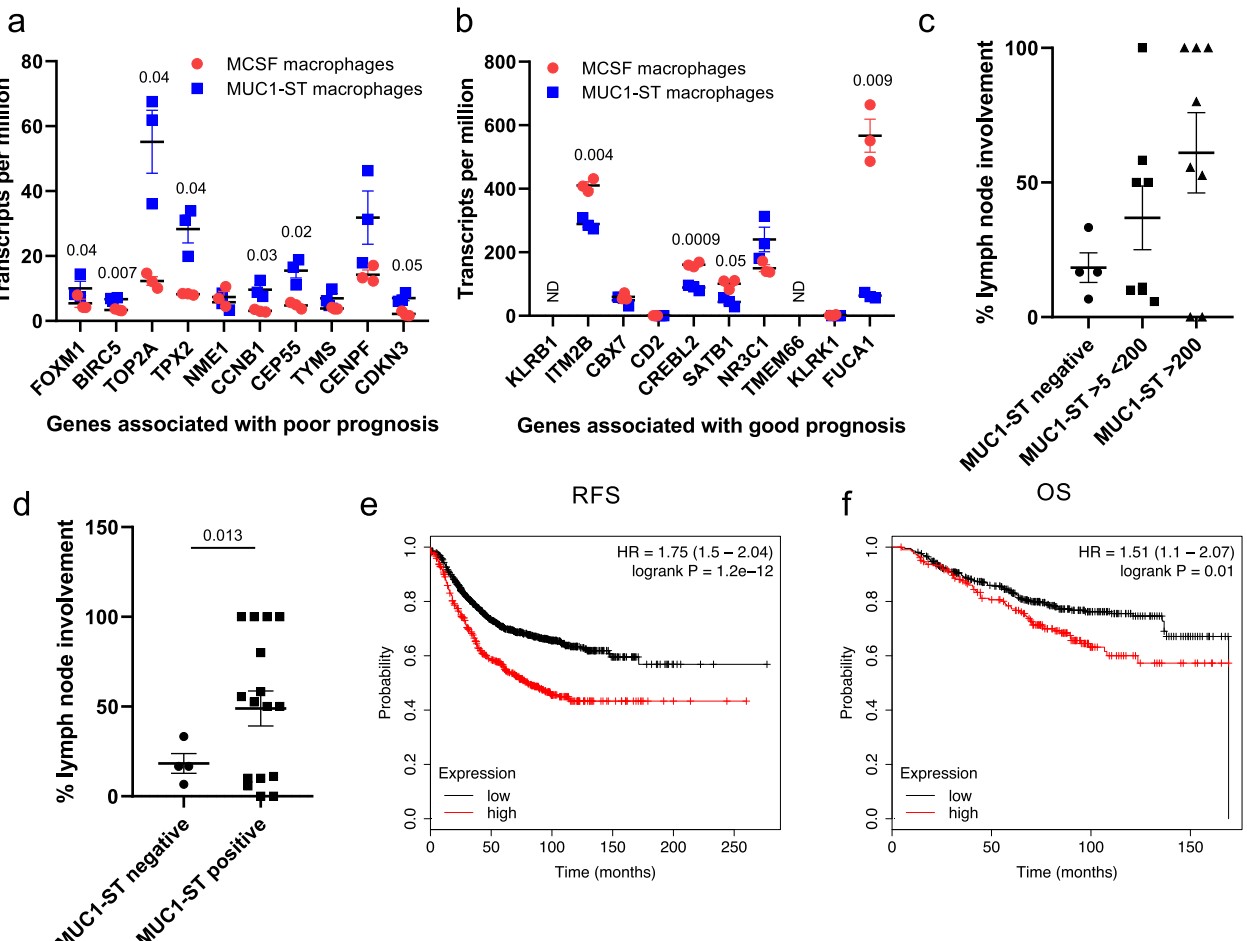

**Fig. 7 MUC1-ST differentially expressed genes are associated with poor clinical outcome. a** Top ten genes associated with a poor prognosis or **b** good prognosis (described in Gentles et al.[25]), expressed by MUC1-ST macrophages (n = 3 biologically independent samples) or MCSF macrophages (n = 3 biologically independent samples). **c, d** Percentage of lymph nodes positive for cancer in relation to their primary cancer MUC1-ST score (n = 20 breast cancer cases), **c** 3 groups, **d** 2 groups. **e, f** A 9 gene signature was derived from top genes (fold change) from the differential expression analysis of MCSF macrophages and MUC1-ST macrophages (fold change > 2, p value >1[10]), and was applied to the BRCA TCGA RNAseq database to generate Kaplan Meier survival curves (Upper third; high signature expression. Lower two thirds; low signature expression). **e** Relapse free survival (RFS); **f** overall survival (OS). ND not detected. Standard errors of the mean shown. **a, b** Statistical analysis using paired t test. **c** Statistical analysis using unpaired t test with Welch's correction owing to unequal population variance.

with a poor prognosis[47,48]. Furthermore, Richardson et al.[48] report that stromal cells expressing M-CSF, also expressed by MUC1-ST-induced macrophages, are associated with metastasis. Importantly we found a correlation between the intensity of MUC1-ST staining of the cancer cells and CD163+ macrophages in the stroma around the nests of cancer cells. Moreover, macrophages with a MUC1-ST-induced phenotype, demonstrated by expression of CXCL5 and *SERPINE1* found in the stroma around the edge of the nests, correlated with MUC1-ST expression. These data suggest that MUC1-ST is driving the generation of these specific TAMs in this specific location.

The aberrant glycosylation of MUC1 is found many in carcinomas as is the upregulation of ST3Gal-1[49]. Therefore, it is likely that a similar mechanism as described here could be occurring in other cancers. The glycosylation of the colon where core 3 O-linked glycans dominate is quite different to the breast. However, the glycosyltransferase responsible for the formation of this core is dramatically downregulated in colon cancer leading to the expression of core 1 glycans[50]. Moreover, staining of the KL-6 antibody that reacts with sialylated MUC1 has been observed in colorectal cancer suggesting the possibility that MUC1-ST could be present in colorectal cancer[51]. However, the role of TAMs in

colorectal cancer is unclear as there are conflicting studies as to their function in this tumour type[52].

Historically, TAMs within human breast cancer had been identified only by immune histochemistry. However, recently macrophages isolated from breast cancers have been analysis by RNAseq[7], CyTOF[53] and single-cell RNAseq[8]. The Pollard lab identified a TAM signature also associated with poor prognosis and that is enriched in HER2 positive breast cancers. One of the identified genes was *SIGLEC1*, which when transcribed and translated engages with CCL8 in a tumour cell regulatory loop[54]. This TAM type is different to the one we have identified as *SIGLEC1* was one of the most highly downregulated genes in the MUC1-ST-induced macrophages (Supplementary Data 1). Comparative and correlative analysis of the transcripts expressed by the MUC1-ST-induced macrophages suggests that the MUC1-ST macrophage subtype is most closely related to subtype 23 identified by Azizi et al.[8]. Interestingly, the authors determined that the TAMs in cluster 23 were of mixed classical 'M1' and 'M2' signatures, something that is apparent in the phenotype of MUC1-ST-induced macrophages.

MUC1-ST-induced macrophages can produce factors that are able to modulate the immune microenvironment. Firstly, factors

such as CXCL5, CXCL8, CCL24, S100A8[28,30] and ALOX5[34] expressed by MUC1-ST-induced macrophages are involved in neutrophil recruitment and our in vitro data show that MUC1-ST-induced macrophages can indeed induce neutrophil migration and also promote neutrophil viability. Increased neutrophil numbers in breast cancers is associated with worse survival[29] and the absence of neutrophils profoundly reduces pulmonary metastasis in a murine model of breast cancer[27]. Conversely, in murine models of mammary cancer neutrophils are associate with good prognosis. Through MET/HGF signalling neutrophils can release nitric oxide which promotes cancer killing and inhibits metastasis[31] and when neutrophils come into contact with tumour cells anti-tumour cytotoxicity is mediated through $H_2O_2$-dependent calcium channel, TRPM2[33].

The factors released by the MUC1-ST-induced macrophages and the functional data suggests a very strong relationship between these macrophages and neutrophils, however further work is required to establish whether this relationship helps or hinders tumour growth and spread.

Secondly, MUC1-ST-induced macrophages produce factors including PD-L1 (CD274), PD-L2 (PDCDILG2), IDO1 and arginase that negatively regulate the activity of T cells, CCL24 which acts to recruit resting T cells but not activated T cells[55], and CCL18 that recruits Tregs[56], whilst also downregulating CD86, important for the co-stimulation of T cells. Interestingly TAMs isolated from breast cancers have previously been seen to secrete large amounts of CCL18 and promote metastasis through CCL18 binding to PITPNM3[57]. Our in vitro studies confirm that MUC1-ST-induced macrophages inhibit the proliferation of T cells and decrease their viability.

MUC1-ST-induced macrophages show further characteristics of TAMs in that they are inefficient at phagocytosis and induce the invasion of both neutrophils and the minimally invasive breast cancer cell line MCF-7. Interestingly the proteases that are upregulated in the MUC1-ST-induced macrophages are MMP14 and MMP2, while MMP9 and MMP19 are downregulated compared to M-CSF-induced macrophages. The inhibitors of MMPs, the TIMPs, are downregulated in MUC1-ST-induced macrophages. MMP14 and MMP2 both degrade the extracellular matrix especially collagen IV, found in basement membranes, and indeed MMP14 and MMP2 have been shown to promote cancer invasion and metastasis[58]. Furthermore, MMP14 can also induce HIF transcription factors independently of its protease activity[59]. Taken together, MUC1-ST-induced macrophages appear to display a combination of MMPs and TIMPs that enable specific degradation of collagen type IV and may explain why the supernatant from MUC1-ST induced macrophages was so potent in our basement membrane extract in vitro invasion assays. It is this basement membrane degradation that has been proposed as a mechanism whereby tumours invade; macrophages or neutrophils 'burrow' towards the tumour allowing cancer cells to escape[60,61].

Patients with cancer are at an increased risk of developing venous thromboembolism often known as Trousseau's syndrome[62]. Although a number of mechanisms have been suggested to modulate thrombogenesis in cancer[63], tissue factor which is the activator of coagulation in vivo, is elevated in the circulation of cancer patients and correlated with mortality[64]. Trousseau's syndrome is associated with mucin-producing adenocarcinomas and may be triggered by the interaction of circulating mucins with P- and L-selectin[65]. Here, we show that MUC1-ST-induced macrophages express factors that are associated with clotting and the secretion of tissue factor (F3 gene) is significantly increased in MUC1-ST-induced macrophages compared to M-CSF. Indeed, our functional studies show that conditioned medium from MUC1-ST-induced macrophages induces faster clotting than medium from M-CSF macrophages.

It is additionally interesting to note that 14/18 of the upregulated protein-validated factors are associated with poor prognosis or invasion in breast cancers, when measured in serum or tissue; this suggests that cells which secrete these factors would have a negative impact on prognosis[39,48,57,58,64,66–82]. It is also of interest to note that in these studies 8/14 of these poor-prognostic factors have been seen to derive predominantly from stromal cells.

The overlap with the Gentles top genes associated with poor prognosis is also striking and it is important to note that these genes are correlated with prognosis in all cancers. As MUC1 is expressed by the vast majority of solid tumours[83], and aberrant hypersialylation is very common, it leaves open the possibility that MUC1-ST-induced macrophages may also present in other carcinomas.

Considering the factors over-expressed by MUC1-ST-induced macrophages, their functionality, transcriptome and location, it is highly likely these cells are pathogenic in breast cancer. Understanding the mechanism by which these cells are produced, in depth, is imperative and may lead to additional targeting opportunities. Indeed, targeting and depleting TAMs is now being evaluated as a potential therapeutic approach[6,84,85] and reprogramming the phenotype of TAMs by the use of HDAC inhibitors and TLR agonists is also being trialled[86,87]. However, TAMs are a heterogeneous group of cells[7–9] and increased knowledge of the large number of subtypes is necessary to make these targeting strategies a success. The presence of MUC1-ST TAMs in primary breast cancers, a MUC1-ST TAM signature being associated with poor prognosis and its phenotype contributing to systemic features of cancer, suggest that approaches based on targeting TAMs should include this subtype. Finally, as MUC1-ST-induced macrophages are induced through interaction with Siglec-9 on monocytes, targeting the Siglec9/MUC1-ST interaction could effectively inhibit the production of the pro-cancer MUC1-ST-induced macrophages and impact on survival[88].

## Methods

**Generation of MUC1 glycoforms.** Recombinant secreted MUC1 consisting of 16 tandem repeats carrying sialylated core 1 and fused to mouse Ig was produced in CHO cells as described in Backstrom et al.[89] and Link et al.[90]. Concentrated supernatant was treated with 10 mg trypsin per mg MUC1-ST-IgG for 2 h (MUC1 tandem repeats are not sensitive to trypsin digestion) to remove the Ig. The treated supernatant was applied to a HiPrep 16/10 Q FF anion exchange column, which was washed to remove the unbound material with 20 column volumes of 50 mM Tris-HCl pH 8.0. The MUC1-ST was eluted as described in Backstrom et al.[89] Quality control procedures include endotoxin testing (LAL), casein cleavage assay, MUC1-lectin ELISA, amino acid analysis and TGFβ1 ELISA on the products. There are additional functional endotoxin controls of (a) TNFα measurement in supernatant of monocytes treated with MUC1-ST or MUC1-T for 48 h, and (b) assessment of readouts after inhibition of NFκB, AP1 and TLR4 pathways.

**Isolation of monocytes.** Leucocyte cones were ordered from the National Health Service Blood and Transplant Service (NHSBTS) (The NHSBTS obtains informed consent from the donors and has internal ethical approval under the terms of HTA licence). Cells were mixed 1:1 with phosphate-buffered saline (PBS) and layered on Ficoll–Paque (GE Healthcare; 1714402). Cells were spun at 800 G for 30 min, with the brake off, and the PBMCs were taken from the buffy layer above the Ficoll–Paque. CD14+ cells were isolated from PBMCs using the MACS system (Miltenyi Biotech; 130-050-201. LS Columns; 130-042-401). Purity was checked using anti-CD14 antibodies (Supplementary Table 1, concentration as per manufacturer's instructions) and seen to be >95%. If purity was below 95%, the cells were disposed of.

**Culture of monocyte-derived macrophages.** Freshly isolated monocytes, from fresh leucocyte cones, were cultured for 7 days at $1 \times 10^6$/ml in AIM-V media (ThermoFisher; 12055091), in the presence of 50 ng/ml recombinant M-CSF (replenished every 3 days; biolegend; 574804) or 25 μg/ml recombinant MUC1-T or MUC1-ST unless otherwise stated in the figures. Cells were counted using a haemocytometer and viability was assessed using a viability dye (ThermoFisher; L23102) and flow cytometry. For M-CSF blocking studies, 10 μg/ml αM-CSF or

isotope control was added every 3 days throughout the culture period. Supernatant was taken from these cells and aliquoted and stored at −20 °C prior to use for functional assays. Bright field images were captured using an EVOS XL Core Cell Imaging System.

**Immunohistochemical staining of MUC1-ST**. As no antibodies are available that specifically react with MUC1-ST we used the 1B9 antibody that binds to MUC1-T with or without treatment of the section with neuraminidase. The protocol was as described[91]. Briefly, 5 µm FFPE sections were dewaxed, blocked with 20% fetal bovine serum (FBS) in PBS for 1 h, before being treated in neuraminidase buffer (50 mM sodium acetate pH5.5) ± neuraminidase (Sigma; N2876; 10 mU/section) for 1 h at 37 °C. Sections were stained using the anti MUC1-T antibody (1B9)[92] for 1 h (neat supernatant), washed twice in PBS, before a secondary (goat anti-mouse HRP; 1:100) was added for 1 h. Sections were washed four times then stained with DAB (Agilent; K3467) and counterstained with haematoxylin. Sections were scanned using a Hammamatsu slide scanner and visualised for scoring using NDP View software (2.7.25). MUC1-ST scoring was determined by subtracting the MUC1-T score (1B9 staining without neuraminidase treatment) from the MUC1-ST score (1B9 staining with neuraminidase treatment) in matched sequential sections. All breast cancer sections were obtained from the King's Health Partners' Tissue Bank under ethical approval obtained by the Bank (East of England—Cambridge Research East Ethics Committee, REC reference 18/EE/0025). All patients gave informed consent for their samples to be used for cancer research.

**Flow cytometry**. Totally, $1 \times 10^5$ cells were stained with a live/dead dye (ThermoFisher; L23102) in PBS for 10 min on ice in the dark, before being washed twice in FACS buffer (0.5% bovine serum albumin [Sigma; 05482] in PBS + 2 mM EDTA). Cells were then Fc blocked with Trustain (Biolegend; 422302) in FACS buffer for 10 min on ice in the dark. Cells were washed and then stained using a variety of antibodies ± secondary reagents described in Supplementary Table 1, using concentrations recommended by the manufacturer, on ice for 30 min in the dark (if secondaries were used, the cells were washed in FACS buffer before being further incubated on ice with secondary, using concentrations recommended by the supplier, for 30 min). Cells were washed and either read immediately or fixed using 1% PFA in FACS buffer and read within 3 days. Cells were read using a BD Accuri C6 Plus flow cytometer, with analysis carried out using BD Accuri C6 Plus software. All cells were gated as follows: (a) Forward scatter and side scatter (SSC) to exclude cellular debris (whilst also adjusting threshold), (b) live/dead (only live cells carried forward) and (c) SSC-A vs. SSC-H—only singlets carried forward. All MFIs were corrected against an appropriate isotype control. Intracellular flow cytometry was carried out using the intracellular fixation and permeabilization kit (ebioscience; 88-8824-00) according to manufacturer's instructions.

**RNAseq library preparation**. Monocytes from three donors were isolated. Matched M-CSF and MUC1-ST monocyte-derived macrophages were cultured as described. Cells were harvested and FACS sorted (BD FACSAria II Cell Sorter) for live cells after staining with a live/dead dye (ThermoFisher; L23102). Total RNA was isolated from the sorted live cells using the RNeasy Mini Kit (Qiagen; 74104) with DNAse treatment (Sigma; DN25). RNA was quantitated using the Qubit system and the RIN score was assessed using an Agilent bioanalyser 2100 (Agilent RNA 6000 Nano Kit). All samples in this study had RIN scores of 10. PolyA isolation and library preparation was performed using SureSelect Strand Specific RNA-Seq Library Preparation kit (G9691B) on 335 ng of RNA per sample. Samples were run on the Illumina platform (HiSeq2500 Rapid) for 25 cycles. All data are deposited in GEO, reference GSE150613.

**RNAseq analysis**. RNA seq analysis was performed on Partek Flow Software (https://www.partek.com/partek-flow/). All the tools with in the software was run with default settings, unless otherwise indicated. The quality of the sequencing reads was examined using FastQC (v0.11.4) (https://www.bioinformatics.babraham.ac.uk/projects/fastqc/). Raw sequencing reads (100 nt, paired-end) were trimmed using Trimgalore (v0.4.4) (https://www.bioinformatics.babraham.ac.uk/projects/trim_galore/). Traces of ribosomal DNA and mitochondrial DNA were removed using the Bowtie2 (v2.2.5)[93]. Reads were aligned to the human reference genome GRCh38 using STAR (v2.5.3a)[94] with two pass mapping multi-sample setting. Mapping and alignment quality were examined using FASTQC. Duplicate reads were removed using the MarkDuplicates function of the Picard tools (v2.17.11) (http://broadinstitute.github.io/picard/). Reads were annotated using the Partek E/M with GENCODE V30 (https://www.gencodegenes.org/human/). Samples were visualised and explored using unsupervised methods. All samples were clustered based on principle component analysis, K-means clustering, tSNE and hierarchical clustering. Gene counts were normalised using the trimmed mean of $M$-values and differentially expressed genes (DEG) between MUC1-ST and M-CSF treated samples were identified using Partek differential expression (DE) analysis tool. DEG with |fold change| ≥ 2 and FDR value ≤ 0.01 were used for pathway enrichment and gene ontology (GO) analysis. GO and pathway enrichment analysis was done using DAVID Bioinformatics Resources 6.8 (https://david.ncifcrf.gov/).

**CIBERSORT analysis**. The CIBERSORT R source code and the LM22 signature matrix file, which defines 22 immune cell types based on the expression levels of 547 genes, were downloaded from https://cibersort.stanford.edu/. Cell type deconvolution was carried out using the default parameters

**ELISA**. CXCL5 (biolegend; 440904) MMP14 (Bio-techne; DY918-05) and Tissue factor (Bio-techne; DY2339) sandwich ELISAs were performed as per manufacturer's instructions. Plates were read on a CLARIOstar instrument at 450 nm, being corrected against 570 nm, and analysed using MARS software and excel. For Siglec-9 blocking studies, monocytes were preincubated with 10 µg/ml αSiglec-9 antibodies or isotype control on ice for 30 min, washed, then incubated with recombinant MUC1-ST for 4 h before being washed and cultured, as per Beatson et al.[25,22].

**Luminex**. Choice of analytes was determined by RNAseq analysis. The Luminex kit was manufactured by Bio-techne and the assay was performed as per manufacturer's instructions. Samples were analysed using Luminex Flexmap3D apparatus and analysis was performed using Xponent 4.0 software. For Siglec-9 blocking studies, monocytes were preincubated with 10 µg/ml αSiglec-9 antibodies or isotype control on ice for 30 min, washed, then incubated with recombinant MUC1-ST for 4 h before being washed and cultured, as per Beatson et al.[22].

**Cell lines**. T47D, MCF7 and E2J (T47D cells, transfected with C2GnT1[26]; T47D (core 2)) cell lines were cultured in DMEM (ThermoFisher; 41966-029) + 10% FBS (ThermoFisher; 10270106) + pen/strep (Sigma; P4333) + glutamax (Thermo-Fisher; 35050-038). E2J cells were selected throughout in 500 µg/ml G418 (Sigma; 04727878001). MCF-7 were authenticated by LGF Standards using short-tandem repeat profiling. E2J and T47D cells have recently been glycophenotyped by mass spectrometry[95]. T47D and MCF-7 were obtained from their originators and all cell lines were regulated tested for mycoplasma and kept in culture for no longer than 3 months. For co-culture experiments cells were cultured in 24 well plates at $1 \times 10^5$/ml the day before the assay. For neuraminidase treatment, culture supernatant was removed, and cells were treated with 40 mU/ml neuraminidase in PBS, or PBS as control, for 30 min at 37 °C, before being gently washed twice with PBS. Successful treatment was visualised by flow cytometry of treated cells; PNA staining (1 µg/ml) increases. Epithelial cells plus monocytes were cultured in AIM-V media for 48 h before supernatant was collected for protein analysis.

**Isolation of neutrophils**. Totally, 4 ml fresh donor blood was taken (REC09/H0804/92) and mixed with 45 µl of sterile 0.5 M EDTA. Neutrophils were isolated from fresh donor blood using MACSexpress whole blood neutrophil isolation kit (Miltenyi; 130-104-434). Eythrocytes were lysed very gently (biolegend; 420301). Purity was assessed to be >95% using CD16, CD15 and CD66b antibodies by flow cytometry (Supplementary Table 1) with manufacturer's recommended concentrations used.

**Migration assay**. Cells were assayed in Bowden chambers with an 8 µm pore size (353097). Freshly isolated neutrophils were placed in the top chamber (150 µl at $1 \times 10^6$/ml in AIMV media). Totally, 650 µl of M-CSF or MUC1-ST macrophage supernatant was placed in the bottom chamber. Migrated cells were counted in the bottom chamber using a haemocytometer at indicated time points, in triplicate.

**Invasion assay**. Cells were assayed in Bowden chambers (353097) layered with extracellular matrix (Sigma; 126–2.5 or Biotechne; 3433-005-01) as per manufacturer's instructions (AIM-V media used to mix). Freshly isolated neutrophils or MCF7 cells were placed in the top chamber (150 µl at $1 \times 10^6$/ml in AIMV media). In total, 650 µl of M-CSF or MUC1-ST macrophage supernatant was placed in the bottom chamber. Migrated cells were counted in the bottom chamber using a haemocytometer at indicated time points, in triplicate.

**Clotting assay**. A 50 µl of human plasma (Sigma; P9523) was added to 50 µl of supernatant from matched M-CSF or MUC1-ST-induced macrophages. A 50 µl of rabbit thromboplastin (Sigma; 44213) was added as a positive control. A 50 µl of 30 mM $CaCl_2$ was added and the optical density was immediately read at 405 on a CARIOstar plate reader as a measure of clotting density as per Ashour et al.[96]. Visual checks were made at the end of the assay. Reads were made every 20 s for 11 min. Data were analysed using MARS software, excel and GraphPad.

**Phagocytosis assays**. T47D cells were labelled with CSFE as per manufacturer's instructions (ebioscience; 65-0850-84), washed three times in media with serum, and co-cultured at a 1:1 ratio with M-CSF and MUC1-ST-induced macrophages for 4 h at 37 and 4 °C. For the dextran work, dextran-FITC (Sigma; FD40S) was added at 1 mg/ml to M-CSF and MUC1-ST-induced macrophages for 4 h at 37 and 4 °C. Cells were analysed by flow cytometry for evidence of uptake. Active phagocytosis was inferred to be the difference between binding (assay at 4 °C) and uptake (assay at 37 °C).

**MLR and plate bound aCD3 assays**. M-CSF or MUC1-ST monocyte-derived macrophages were generated as described. Allogeneic PBMCs were stained with CFSE proliferation dye as per manufacturer's instructions and co-cultured at a 1:5 ratio (mφ:PBMC) with monocyte-derived macrophages. Cells were cultured for 4 days before being assessed for daughter populations by flow cytometry. For the αCD3 assays, 96 well flat-bottomed tissue culture plates were coated with 1 μg/ml αCD3 overnight at 4 °C. Plates were washed with PBS, and PBMCs, pre-stained with efluor670 proliferation dye as per manufacturer's instructions, were added along with supernatant from MUC1-ST-induced macrophages or media alone. Cells were cultured for 4 days before being assessed for daughter populations by flow cytometry.

**Ventana staining**. Sections were stained for CD163 and CD68 using the Ventana Benchmark Ultra system using Ventana pre-diluted antibodies and standard CC1 with the benchmark Ultraview DAB detection kit. Positive control sections were run with every batch.

**Tissue scoring: CD163, SERPINE1, CSF1 and CXCL5+CD68+ scoring**. These were scored by 5 individuals using a 0–4 scoring system as follows:

Score 0 = negative; score 1 = 0–5% positive cells; score 2 = 5–20% positive cells; score 3 = 20–60% positive cells; score 5 = 60–100% positive cells.

CD163 was taken forward for the Visiopharm analysis and chromogenic scoring. CD68 was included for immunofluorescent staining as the differential between background and positive staining was excellent.

**MUC1-ST scoring**. To provide greater scoring sensitivity for correlation analysis the product of percentage coverage (0–100) and intensity (0–5) was recorded for each case. These scores were performed by three individuals.

Geographical regions.

- *Nest*. Positive cells integrated within the tumour.
- *Edge of nest*. Positive cells on the edge of the tumour; from 0, i.e. touching tumour cells on the outer edge, to 200 μm.
- *Stroma*. Positive cells beyond 200 μm from edge of tumour.

**RNAscope**. RNAscope using the duplex system was carried out as per manufacturer's instructions using the manual method (Biotechne; 322430). Hs-SERPINE1 (Biotechne; 555961) and Hs-CSF1 (Biotechne; 313001-C2) probes were used.

**BASEscope**. BASEscope using the duplex system was carried out as per manufacturer's instructions using the manual method (Biotechne; 323810). BA-Hs-SIGLEC9-tv2-1zz-st, which binds to SIGLEC9 transcript variants 1 and 2, was designed by Bio-techne and used.

**Immunofluorescent immunohistochemistry**. Totally, 5 μm FFPE sections were dewaxed, treated with $H_2O_2$ before performing antigen retrieval. Sections were boiled in citrate buffer (Sigma; C9999) for 30 min. Sections were washed in PBS Tween, then blocked 50% FBS for 1 h. After washing, sections were probed with anti CD68 (1:100) and anti CXCL5 (1:50) for 1 h. After further washing, sections were stained with donkey anti-mouse 488 (1:1000) and donkey anti goat 557 (1:200) in 10% FBS and incubated for 1 h. Final washes were performed, and sections were stained with DAPI for 30 s before being mounted (Vector Labs; H-100). Sections were scanned using an Olympus BX61VS and images were analysed using OylVIA software.

**Visiopharm (digital pathology analysis software)**. NDP (Hammamatsu) images were analysed using VisioPharm analysis software. Briefly, images of CD163 stained slides were segmented into tumour vs non-tumour by creating an Application Protocol Package (APP) in the Visiopharm software, training the DeepLab v3 algorithm to differentiate between the tumour region of interest (ROI) vs. the non-tumour. Deep learning involves neural network algorithms that use a cascade of many layers of nonlinear processing units for feature extraction and transformation with each successive layer using the output from the previous layer as input. Using deep learning for classification allows to segment abstract image structures that would be impossible to segment with a simple pixel classifier. In particular, DeepLabv3+ uses spatial pyramid pooling (ASPP) module augmented with image-level features to capture feature information on different scales. Post-processing steps were added to remove noise, calculate total area of ROI's, and create a tumour border ROI (300px thick region from tumour ROI into non-tumour ROI). Subsequently, a threshold algorithm-based APP for DAB staining was adjusted and used on the tumour images, to identify the percentage of total area in ROI's expressing CD163. This classification method is based on a custom defined input band, the so called HDAB, which takes haematoxylin and DAB staining into consideration by having the two stains as the primary and secondary axis in the colour space coordinate system.

**Interstitial fluid (ISF) collection**. The method of Celis et al.[97] was followed. Briefly, fresh breast tissue, collected under ethical approval REC number 12/EE/0493, was diced into 1–3 mm³ pieces and incubated for 1 h at 37 °C in 1 ml of PBS. After incubation tissue was spun at 1000 G for 2 min and supernatant removed and spun for a further 20 min at 4 °C at 5000 G. Supernatant (ISF) was removed and stored at −20 °C for subsequent analysis.

**TCGA correlations analyses**. TCGA (BRCA) expression data for genes of interest were analysed and downloaded from xenabrowser.net (University of Santa Cruz).

**Signature generation and application**. The nine gene signature was generated by applying the following filters to the >2 fold change RNAseq differential gene list (Supplementary Data 1, tab 2) and sorting on fold change. Transcripts per million threshold of 10. P value of >10^10. Top nine genes taken independent of z-score.

**Survival analysis**. KMplot (www.kmplot.com)[98] was used to assess the prognostic impact of the MUC1-ST macrophage signature on patient disease and outcome, using the TCGA array and RNAseq datasets. The upper tertile was used to split the high and low populations and only JetSet probes were used.

**Clinical data**. Clinical data was collected, linked and anonymised by the King's Health Partners Tissue Bank. The use of tissue and data from King's Health Partners Cancer Biobank was approved under REC number 12/EE/0493.

**Statistics and reproducibility**. Statistical analysis was performed using GraphPad Prism software or MS excel. Appropriate group analysis tests were determined by assessing number of comparative groups, variance and whether the data was paired or not. Correlation analysis was performed using linear regression analysis (Pearson's). Sample sizes were determined by setting a minimum n number for in vitro biological replicates at 3, to allow for statistical testing, however in most cases n numbers were higher, ranging from 3 to 14. All replicates displayed in this paper are biological replicates, technical replicates (usually 3) were performed and used to generate the means for each biological replicate. For the tissue analysis after applying stringent power calculations, we acquired 60 cases, however, for 7 cases the tissue quality was too poor to analyse. We were blinded to both the pathological and clinical information, being unblinded after analysis was complete.

**Reporting summary**. Further information on research design is available in the Nature Research Reporting Summary linked to this article.

## Data availability

The data related to the RNAseq experiments are deposited in GEO reference GSE150613 and can be found in Supplementary Data 1. Data related to Figs. 1–7 including flow data, ELISA data and Luminex data can be found in Supplementary Data 2. Data relating to Supplementary figures is available from Richard Beatson or Joy Burchell upon reasonable request.

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

## Acknowledgements

This work was supported by MRC grants MR/R000026/1 and MR/J007196/1 and CRUK KHP Centre Grant. We thank to Mansoor Saqi, Ulrich Kaldolsky and Rianne Wester of The National Institute for Health Research Biomedical Research Centre at Guy's and St. Thomas' NHS Foundation Trust; The National Health Service Blood and Transplant Service, in particular Michael Saunders and Julie Stacey, for supplying leucocyte cones from healthy donors; Nicola O'Reilly at the Peptide Synthesis Lab at the CRICK Institute for support and lyophilisation; Katie Flaherty for co-scoring images and Toby Lawrence for helpful discussion. The first author would like to dedicate this work to the memory of Lucy Beatson who passed away during the course of this work.

## Author contributions

R.B. designed the study, conducted the experiments and contributed to the first draft of the paper; R.G. and F.G.F. carried out the experiments; D.C. and S.K. provided the bioinformatics support; R.M. made the RNAseq libraries; N.W., J.O. and C.G. supplied the breast cancer samples; I.B. and S.P. provided help with the VisoPharm; U.M. provided the reagents: T.N. cultured CHO cell transfected with the recombinant MUC1 in serum-free medium; J.T.-P. and E.P. contributed helpful discussion; J.M.B. designed the study and wrote the first draft of the paper. All authors read the paper and made contributions.

## Competing interests

Joy Burchell is a consultant to Palleon Pharmaceuticals. The remaining authors declare no competing interests.
