## [Peer Review File · Communications Biology]

Reviewers' comments:

Reviewer #1 (Remarks to the Author):

Review of manuscript "Cancer-associated hypersialylated MUC1 drives the differentiation of monocytes into macrophages with a pathogenic phenotype" by Richard Beatson, Rosalind Graham, Fabio Grundland Freile, Domenico Cozzetto, Shichina Kannambath, Ester Pfeifer, Natalie Woodman, Julie Owen, Rosamond Nuamah, Ulla Mandel, Sarah Pinder, Cheryl Gillett, Thomas Noll, Ihsane Bouybayoune, Joyce Taylor-Papadimitriou, Joy M. Burchell.

In this manuscript, the authors present a very interesting set of data that establishes that a sialylated form of MUC1 (MUC1-ST) can induce differentiation of monocytes into a novel macrophage subset. The authors provide evidence that these MUC1-ST macrophages are TAM-like and are associated with the edge of tumor nests. Differentiation into MUC1-ST macrophages can be (partially) blocked with anti-Siglec-9 antibodies. The MUC1-ST macrophages have pathogenic effects on neutrophils, T cells, blood clotting, have reduced phagocytosis and express specific MMPs. The presented data are convincing, but the manuscript needs to be improved in different ways. My major concern is with the way the experiments and methods are introduced and described.

Major comments

1. The authors emphasize the involvement of Siglec-9 in their findings. They state that "CXCL5 and SERPINE/PAI-1 are two of the most highly differentially expressed genes in MUC1-ST macrophages". PAI-1 expression is not significantly blocked by the anti-Siglec9 antibody (Figure 3i). Removal of the sialic acid of MUC1-ST leads to significant (?) reduction of PAI-1 expression (Figure 3h). Could other Siglecs be involved in the effect of MUC1-ST?
2. How are the CXCL5 and SERPINE/PAI-1 genes related to Siglec-9 signaling? There is no discussion on how MUC1-ST signaling through Siglec-9 could induce the differentiation of TAM-like macrophages.
3. Where is Siglec-9 in the expression plot in Figure 3C and is it among the genes in Figure 3D and E?
4. The manuscript is not an easy read. It is often challenging to follow the line of logic and argument, especially in the results section. The reader needs to go back and forth between text, methods and figures to grasp the full story. Please rewrite to improve legibility.
5. Reduce extremely long sentences such as "Given that MUC1-ST induced macrophages expressed genes associated with extracellular matrix disassembly, particularly MMP14 the expression of which is dependent on the sialic acid carried on MUC1-ST (figure 4e and 4f), and given the reported importance of macrophage mediated basement membrane degradation in promoting invasion and metastasis 36,37, the invasion of neutrophils through basement membrane extract towards the various supernatants was investigated." (page 7)
6. The experiments described in this manuscript follow from a large body of work published by this group. However, please make sure that it is clear to the (maybe uninformed) reader how experiments are conducted. For example, it is unclear to me how MUC1-ST was stained in Figure 1. Only mention of neuraminidase treatment and MUC1-T antibody in the methods. No mention of MUC1-ST antibody. There is a reference to Beatson et al., 2015 but also in these methods it is not clear how MUC1-ST is stained. How can the authors be sure that they are looking at MUC1-ST signal and not MUC1-T + MUC1-ST (after neuraminidase treatment)? Also make sure that methods are properly introduced in the results section.

Minor comments

1. Abstract – 3x show in 2 sentences
2. In the abstract and on page 9 and 10. I'm not familiar with the term "maintain neutrophils". What does it mean? Does it refer to viability?
3. Figure 1A: please include an example of negative staining

4. In the methods: "neuramindase" should be neuraminidase
5. Figure 1e: in the text it is mentioned that this figure displays counting by manual methodology and Figure S1C by automated methodology. This is not mentioned in the legend of these figures. The methods of these two methodologies are also unclear. Is the automated methodology the Visiopharm? Please make sure that the used methods are clear in the result section and legend and that they are described in detail in the materials and methods.
6. Figure 3C: the labeling of the graph is not clear. What relative expression is depicted in the graph?
7. Figure 7D: it is not clear from the labeling or legend what is depicted in this graph. From the description of the results I gather that we are looking at a MUC1-ST gene signature. Please adjust the legend.
8. Page 10 "shows a dramatic change in glycosylated in many. Should be "glycosylation"

Reviewer #2 (Remarks to the Author):

This manuscript from Beatson et al. describes the ability of recombinant sialated MUC1, through its interaction with Siglec 9 on monocytes, to directly drive the differentiation of the latter into a macrophage phenotype with many of the functional and transcriptional characteristics of tumor associated macrophages or TAMs. The presence of these TAMs is known to correlate with a poor prognosis in breast cancer. Overall these are well-executed studies that are appropriately interpreted.

While a direct side-by-side transcriptomic comparison – using RNA seq of TAMs from freshly isolated tumors and the MUC1 induced TAMs describe here would have been ideal, there is sufficient and convincing data provided in the manuscript at this time to convince this reader that the data is interesting and potentially relevant.

Minor comments:

P 3: "MUC 1 is expressed on the cell surface or secreted". Is "secreted" correct? Is it not mainly cleaved by TACE and MMP-MT1?

P 12: Please change "which when translated" to "which when transcribed and translated". The authors are aware that genes cannot be directly translated

Discussion: "MUC1-ST induced macrophages" and "MUC1-ST macrophages" are used interchangeably. The later terminology may be confusing and is probably best avoided. It may be taken to mean that these macrophages make MUC1 while others do not.

Discussion: Please briefly discuss relevance of lack thereof of these findings to other epithelial cancers, including colon cancer, in which MUC1 is overexpressed, paying attention to known differences in MUC1 glycosylation.

Reviewer #3 (Remarks to the Author):

In their manuscript entitled "Cancer-associated hypersialylated MUC1 drives the differentiation of monocytes into macrophages with a pathogenic phenotype", Beatson et al. show that a sialylated glycoform of MUC1 (MUC1-ST) promotes monocyte differentiation into tumor-associated macrophages (TAMs). This effect is mediated by binding to Siglec-9, a member of the family of sialic acid binding lectins, mainly expressed by immune cells such as monocytes and macrophages.

The manuscript builds on a previous publication by the same group (Beatson et al. Nat. Immunol. 2016, 17, 1273-81), in which the authors demonstrated MUC1 binding and modulation of the tumor immunological microenvironment through interaction with Siglec-9. The present manuscript, however, markedly extends the previous data, for instance by showing the impact of MUC1-ST macrophages on neutrophil functions, clotting, or T cell activation. Interestingly, the authors were also able to correlate the transcriptional profile of the MUC-ST macrophages with poor prognosis/clinical outcome. Thus, the manuscript is highly original and provides in-depth insights into the functional role of the MUC1-ST/Siglec-9 interaction in monocyte differentiation into TAMs with a unique phenotype. Experiments are well controlled and a wealth of data supports the conclusions drawn by the authors. The manuscript is excellently written and relevant studies are cited and discussed in the Discussion section. I have only minor suggestions for improvement (see Specific points below).

Specific points:

- 1.) In some cases, the rationale for focusing on specific differentially expressed genes in the MUC1-ST macrophages is not fully evident. For instance, in Figure 3f, authors focused on CXCL5 and SERPINE/PAI-1 as two of the highly expressed genes. Which selection criteria did the authors use to in- or exclude differentially expressed genes in further analyses? Thus, in some instances, the rationale for the selection of the respective genes could be described more in detail in the manuscript.
- 2.) As far as I can see, neutrophil viability was assessed by measuring size and granularity (FSC, SSC) using flow cytometry (Figures 4 b,c). Did the authors include additional read-outs and methods to determine neutrophil viability (such as viability dyes)?
- 3.) In the part on the impact of the MUC1-ST macrophages on neutrophil functions (Figure 4 and Discussion), the authors mainly focus on pro-tumor neutrophil functions. Indeed, numerous recent studies show that neutrophils play major roles in linking inflammation and cancer and contribute to tumor progression and metastasis. However, neutrophils can also exhibit anti-tumor functions such as cytotoxicity or the activation of adaptive immune responses. In this regard, data shown in Figure 4 should be interpreted with care. The authors should discuss the ambivalent role of neutrophils in cancer more in detail.

Response to the reviewers' comments on the manuscript Beatson et al.

Reviewer #1 (Remarks to the Author):

In this manuscript, the authors present a very interesting set of data that establishes that a sialylated form of MUC1 (MUC1-ST) can induce differentiation of monocytes into a novel macrophage subset. The authors provide evidence that these MUC1-ST macrophages are TAM-like and are associated with the edge of tumor nests. Differentiation into MUC1-ST macrophages can be (partially) blocked with anti-Siglec-9 antibodies. The MUC1-ST macrophages have pathogenic effects on neutrophils, T cells, blood clotting, have reduced phagocytosis and express specific MMPs. The presented data are convincing, but the manuscript needs to be improved in different ways. My major concern is with the way the experiments and methods are introduced and described.

Reply: We thank the reviewer for their positive comments and have replied to each of the points below:

Major comments

1. The authors emphasize the involvement of Siglec-9 in their findings. They state that "CXCL5 and SERPINE/PAI-1 are two of the most highly differentially expressed genes in MUC1-ST macrophages". PAI-1 expression is not significantly blocked by the anti-Siglec9 antibody (Figure 3i). Removal of the sialic acid of MUC1-ST leads to significant (?) reduction of PAI-1 expression (Figure 3h). Could other Siglecs be involved in the effect of MUC1-ST?

Reply: We have now included new data that show that the binding of MUC1-ST to monocytes can be blocked by over 90% using an anti-body to Siglec-9 and show this data in figure S2a and discuss it on page 5. We have also discussed the data on page 12 of the Discussion and state that, "The transcripts of activating Siglecs 14 and 16 are significantly decreased in MUC1-ST induced macrophages, therefore we cannot exclude the possibility that these Siglecs may also have a role in driving these observations. However, given the data that over 90% of the binding of MUC1-ST to monocytes can be inhibited by blocking Siglec-9 (figure S2a) this seems unlikely".

2. How are the CXCL5 and SERPINE/PAI-1 genes related to Siglec-9 signaling? There is no discussion on how MUC1-ST signaling through Siglec-9 could induce the differentiation of TAM-like macrophages.

Reply: We have previously shown that MUC1-ST binding to monocytes did not induce phosphorylation of Siglec-9 or SHP-1 which is associated with inhibitory signalling. In contrast, down-stream activation of the MEK-ERK pathway occurred (Beatson et al Nature Immunology 2016). In the current paper we have now included new experiments that show that inhibiting the MEK/ERK pathway with PD98059 profoundly inhibits the differentiation of monocytes in response to MUC1-ST (new figure S2d). In addition, further new experiments show that CXCL5 and CD206 (MMR) expression was inhibited by the use of a MEK/ERK inhibitor prior to initial stimulation with MUC1-ST and this is shown in figure S3f. The expression of both of these genes at the protein level is at least partially inhibited by the removal of sialic acid and a Siglec-9 antibody (Figure 3h and i). We have also included a paragraph in the Discussion, discussing these results and suggesting ways in which engagement of Siglec-9 could lead to an activation signal.

3. Where is Siglec-9 in the expression plot in Figure 3C and is it among the genes in Figure 3D and E?

Reply: We have included in the expression of Siglec-9 in figure 3C. Siglec-9 was down-regulated in the MUC1-ST induced macrophages although this did not reach significance ($p=0.077$). We have described these results, along with the expression of other siglecs on page 7.

As Siglec-9 was not significantly down-regulated it is not among the genes listed in figure 3E.

4. The manuscript is not an easy read. It is often challenging to follow the line of logic and argument, especially in the results section. The reader needs to go back and forth between text, methods and figures to grasp the full story. Please rewrite to improve legibility.

Reply: We have re-written some of the Results section

5. Reduce extremely long sentences such as “Given that MUC1-ST induced macrophages expressed genes associated with extracellular matrix disassembly, particularly MMP14 the expression of which is dependent on the sialic acid carried on MUC1-ST (figure 4e and 4f), and given the reported importance of macrophage mediated basement membrane degradation in promoting invasion and metastasis 36,37, the invasion of neutrophils through basement membrane extract towards the various supernatants was investigated.” (page 7) .

Reply: The long sentences have been reduced and the sentence quoted by the reviewer changed to, “ MUC1-ST induced macrophages expressed genes associated with extracellular matrix disassembly, particularly MMP14, the expression of which is dependent on the sialic acid carried on MUC1-ST (figure 4e and 4f). As it has been reported that macrophages mediate basement membrane degradation to promote invasion and metastasis ^{30,31}, the invasion of neutrophils and cancer cells through basement membrane extract towards the various supernatants was investigated.”

6. The experiments described in this manuscript follow from a large body of work published by this group. However, please make sure that it is clear to the (maybe uninformed) reader how experiments are conducted. For example, it is unclear to me how MUC1-ST was stained in Figure 1. Only mention of neuraminidase treatment and MUC1-T antibody in the methods. No mention of MUC1-ST antibody. There is a reference to Beatson et al., 2015 but also in these methods it is not clear how MUC1-ST is stained. How can the authors be sure that they are looking at MUC1-ST signal and not MUC1-T + MUC1-ST (after neuraminidase treatment)? Also make sure that methods are properly introduced in the results section.

Reply: We apologise for this and have now made it clear in the Methods how the staining of MUC1-ST was carried out.

Minor comments

1. Abstract – 3x show in 2 sentences

Reply: corrected.

2. In the abstract and on page 9 and 10. I'm not familiar with the term “maintain neutrophils”. What does it mean? Does it refer to viability?

Reply: This has been changed to “prolong the lifespan of neutrophils”

3. Figure 1A: please include an example of negative staining

Reply: Due to the size of the figure we have not included an example of a negative stain but would be very happy to do so if required.

4. In the methods: “neuramindase” should be neuraminidase

Reply: Corrected.

5. Figure 1e: in the text it is mentioned that this figure displays counting by manual methodology and Figure S1C by automated methodology. This is not mentioned in the legend of these figures. The methods of these two methodologies are also unclear. Is the automated methodology the Visiopharm? Please make sure that the used methods are clear in the result section and legend and that they are described in detail in the materials and methods.

Reply: This has been clarified in the Results and figure legend.

6. Figure 3C: the labelling of the graph is not clear. What relative expression is depicted in the graph?

Reply: An addition label has been included.

7. Figure 7D: it is not clear from the labeling or legend what is depicted in this graph. From the description of the results I gather that we are looking at a MUC1-ST gene signature. Please adjust the legend.

Reply: Legend has been adjusted.

8. Page 10 “shows a dramatic change in glycosylated in many. Should be “glycosylation”

Reply: Corrected

Reviewer #2 (Remarks to the Author):

This manuscript from Beatson et al. describes the ability of recombinant sialated MUC1, through its interaction with Siglec 9 on monocytes, to directly drive the differentiation of the latter into a macrophage phenotype with many of the functional and transcriptional characteristics of tumor associated macrophages or TAMs. The presence of these TAMs is known to correlate with a poor prognosis in breast cancer. Overall these are well-executed studies that are appropriately interpreted.

While a direct side-by-side transcriptomic comparison – using RNA seq of TAMs from freshly isolated tumors and the MUC1 induced TAMs describe here would have been ideal, there is sufficient and convincing data provided in the manuscript at this time to convince this reader that the data is interesting and potentially relevant.

Reply: We thank the reviewer for their very positive comments

Minor comments:

P 3: “MUC 1 is expressed on the cell surface or secreted”. Is “secreted” correct? Is it not mainly cleaved by TACE and MMP-MT1?

Reply: This has been corrected, see page 3

P 12: Please change “which when translated” to “which when transcribed and translated”. The authors are aware that genes cannot be directly translated

Reply: Corrected

Discussion: “MUC1-ST induced macrophages” and “MUC1-ST macrophages” are used interchangeably. The later terminology may be confusing and is probably best avoided. It may be taken to mean that these macrophages make MUC1 while others do not.

Reply: Corrected to use MUC1-ST induced macrophages.

Discussion: Please briefly discuss relevance of lack thereof of these findings to other epithelial cancers, including colon cancer, in which MUC1 is overexpressed, paying attention to known differences in MUC1 glycosylation.

Reply: This has been included in the discussion, page 13.

Reviewer #3 (Remarks to the Author):

In their manuscript entitled "Cancer-associated hypersialylated MUC1 drives the differentiation of monocytes into macrophages with a pathogenic phenotype", Beatson et al. show that a sialylated glycoform of MUC1 (MUC1-ST) promotes monocyte differentiation into tumor-associated macrophages (TAMs). This effect is mediated by binding to Siglec-9, a member of the family of sialic acid binding lectins, mainly expressed by immune cells such as monocytes and macrophages. The manuscript builds on a previous publication by the same group (Beatson et al. Nat. Immunol. 2016, 17, 1273-81), in which the authors demonstrated MUC1 binding and modulation of the tumor immunological microenvironment through interaction with Siglec-9. The present manuscript, however, markedly extends the previous data, for instance by showing the impact of MUC1-ST macrophages on neutrophil functions, clotting, or T cell activation. Interestingly, the authors were also able to correlate the transcriptional profile of the MUC-ST macrophages with poor prognosis/clinical outcome. Thus, the manuscript is highly original and provides in-depth insights into the functional role of the MUC1-ST/Siglec-9 interaction in monocyte differentiation into TAMs with a unique phenotype. Experiments are well controlled and a wealth of data supports the conclusions drawn by the authors. The manuscript is excellently written and relevant studies are cited and discussed in the Discussion section. I have only minor suggestions for improvement (see Specific points below).

Reply: We thank the reviewer for their very positive comments.

Specific points:

1.) In some cases, the rationale for focusing on specific differentially expressed genes in the MUC1-ST macrophages is not fully evident. For instance, in Figure 3f, authors focused on CXCL5 and SERPINE/PAI-1 as two of the highly expressed genes. Which selection criteria did the authors use to in- or exclude differentially expressed genes in further analyses? Thus, in some instances, the rationale for the selection of the respective genes could be described more in detail in the manuscript.

Reply: We have described the rationale for selecting CXCL5 and SERPINE/PAI-1 in more detail on page 6.

2.) As far as I can see, neutrophil viability was assessed by measuring size and granularity (FSC, SSC) using flow cytometry (Figures 4 b,c). Did the authors include additional read-outs and methods to determine neutrophil viability (such as viability dyes)?

Reply: Live neutrophils were determined by using a viability dye and this has been made clear in the figure legend.

3.) In the part on the impact of the MUC1-ST macrophages on neutrophil functions (Figure 4 and Discussion), the authors mainly focus on pro-tumor neutrophil functions. Indeed, numerous recent studies show that neutrophils play major roles in linking inflammation and cancer and contribute to tumor progression and metastasis. However, neutrophils can also exhibit anti-tumor functions such as cytotoxicity or the activation of adaptive immune responses. In this regard, data shown in Figure 4 should be interpreted with care. The authors should discuss the ambivalent role of neutrophils in cancer more in detail.

Reply: Extra discussion into the role of neutrophils in cancer has been included and new references given, see page 7 in the results and page 14 in the Discussion.

Reviewers' comments:

Reviewer #1 (Remarks to the Author):

Review of manuscript "Cancer-associated hypersialylated MUC1 drives the differentiation of monocytes into macrophages with a pathogenic phenotype" by Richard Beatson, Rosalind Graham, Fabio Grundland Freile, Domenico Cozzetto, Shichina Kannambath, Ester Pfeifer, Natalie Woodman, Julie Owen, Rosamond Nuamah, Ulla Mandel, Sarah Pinder, Cheryl Gillett, Thomas Noll, Ihsane Bouybayoune, Joyce Taylor-Papadimitriou, Joy M. Burchell.

In this manuscript, the authors present a very interesting set of data that establishes that a sialylated form of MUC1 (MUC1-ST) can induce differentiation of monocytes into a novel macrophage subset. The authors provide evidence that these MUC1-ST macrophages are TAM-like and are associated with the edge of tumor nests. Differentiation into MUC1-ST macrophages can be (partially) blocked with anti-Siglec-9 antibodies. The MUC1-ST macrophages have pathogenic effects on neutrophils, T cells, blood clotting, have reduced phagocytosis and express specific MMPs. The presented data are convincing, but the manuscript needs to be improved in different ways. My major concern is with the way the experiments and methods are introduced and described.

Major comments

1. The authors emphasize the involvement of Siglec-9 in their findings. They state that "CXCL5 and SERPINE/PAI-1 are two of the most highly differentially expressed genes in MUC1-ST macrophages". PAI-1 expression is not significantly blocked by the anti-Siglec9 antibody (Figure 3i). Removal of the sialic acid of MUC1-ST leads to significant (?) reduction of PAI-1 expression (Figure 3h). Could other Siglecs be involved in the effect of MUC1-ST?
2. How are the CXCL5 and SERPINE/PAI-1 genes related to Siglec-9 signaling? There is no discussion on how MUC1-ST signaling through Siglec-9 could induce the differentiation of TAM-like macrophages.
3. Where is Siglec-9 in the expression plot in Figure 3C and is it among the genes in Figure 3D and E?
4. The manuscript is not an easy read. It is often challenging to follow the line of logic and argument, especially in the results section. The reader needs to go back and forth between text, methods and figures to grasp the full story. Please rewrite to improve legibility.
5. Reduce extremely long sentences such as "Given that MUC1-ST induced macrophages expressed genes associated with extracellular matrix disassembly, particularly MMP14 the expression of which is dependent on the sialic acid carried on MUC1-ST (figure 4e and 4f), and given the reported importance of macrophage mediated basement membrane degradation in promoting invasion and metastasis 36,37, the invasion of neutrophils through basement membrane extract towards the various supernatants was investigated." (page 7)
6. The experiments described in this manuscript follow from a large body of work published by this group. However, please make sure that it is clear to the (maybe uninformed) reader how experiments are conducted. For example, it is unclear to me how MUC1-ST was stained in Figure 1. Only mention of neuraminidase treatment and MUC1-T antibody in the methods. No mention of MUC1-ST antibody. There is a reference to Beatson et al., 2015 but also in these methods it is not clear how MUC1-ST is stained. How can the authors be sure that they are looking at MUC1-ST signal and not MUC1-T + MUC1-ST (after neuraminidase treatment)? Also make sure that methods are properly introduced in the results section.

Minor comments

1. Abstract – 3x show in 2 sentences
2. In the abstract and on page 9 and 10. I'm not familiar with the term "maintain neutrophils". What does it mean? Does it refer to viability?
3. Figure 1A: please include an example of negative staining

4. In the methods: "neuramindase" should be neuraminidase
5. Figure 1e: in the text it is mentioned that this figure displays counting by manual methodology and Figure S1C by automated methodology. This is not mentioned in the legend of these figures. The methods of these two methodologies are also unclear. Is the automated methodology the Visiopharm? Please make sure that the used methods are clear in the result section and legend and that they are described in detail in the materials and methods.
6. Figure 3C: the labeling of the graph is not clear. What relative expression is depicted in the graph?
7. Figure 7D: it is not clear from the labeling or legend what is depicted in this graph. From the description of the results I gather that we are looking at a MUC1-ST gene signature. Please adjust the legend.
8. Page 10 "shows a dramatic change in glycosylated in many. Should be "glycosylation"

Reviewer #2 (Remarks to the Author):

This manuscript from Beatson et al. describes the ability of recombinant sialated MUC1, through its interaction with Siglec 9 on monocytes, to directly drive the differentiation of the latter into a macrophage phenotype with many of the functional and transcriptional characteristics of tumor associated macrophages or TAMs. The presence of these TAMs is known to correlate with a poor prognosis in breast cancer. Overall these are well-executed studies that are appropriately interpreted.

While a direct side-by-side transcriptomic comparison – using RNA seq of TAMs from freshly isolated tumors and the MUC1 induced TAMs describe here would have been ideal, there is sufficient and convincing data provided in the manuscript at this time to convince this reader that the data is interesting and potentially relevant.

Minor comments:

P 3: "MUC 1 is expressed on the cell surface or secreted". Is "secreted" correct? Is it not mainly cleaved by TACE and MMP-MT1?

P 12: Please change "which when translated" to "which when transcribed and translated". The authors are aware that genes cannot be directly translated

Discussion: "MUC1-ST induced macrophages" and "MUC1-ST macrophages" are used interchangeably. The later terminology may be confusing and is probably best avoided. It may be taken to mean that these macrophages make MUC1 while others do not.

Discussion: Please briefly discuss relevance of lack thereof of these findings to other epithelial cancers, including colon cancer, in which MUC1 is overexpressed, paying attention to known differences in MUC1 glycosylation.

Reviewer #3 (Remarks to the Author):

In their manuscript entitled "Cancer-associated hypersialylated MUC1 drives the differentiation of monocytes into macrophages with a pathogenic phenotype", Beatson et al. show that a sialylated glycoform of MUC1 (MUC1-ST) promotes monocyte differentiation into tumor-associated macrophages (TAMs). This effect is mediated by binding to Siglec-9, a member of the family of sialic acid binding lectins, mainly expressed by immune cells such as monocytes and macrophages.

The manuscript builds on a previous publication by the same group (Beatson et al. Nat. Immunol. 2016, 17, 1273-81), in which the authors demonstrated MUC1 binding and modulation of the tumor immunological microenvironment through interaction with Siglec-9. The present manuscript, however, markedly extends the previous data, for instance by showing the impact of MUC1-ST macrophages on neutrophil functions, clotting, or T cell activation. Interestingly, the authors were also able to correlate the transcriptional profile of the MUC-ST macrophages with poor prognosis/clinical outcome. Thus, the manuscript is highly original and provides in-depth insights into the functional role of the MUC1-ST/Siglec-9 interaction in monocyte differentiation into TAMs with a unique phenotype. Experiments are well controlled and a wealth of data supports the conclusions drawn by the authors. The manuscript is excellently written and relevant studies are cited and discussed in the Discussion section. I have only minor suggestions for improvement (see Specific points below).

Specific points:

- 1.) In some cases, the rationale for focusing on specific differentially expressed genes in the MUC1-ST macrophages is not fully evident. For instance, in Figure 3f, authors focused on CXCL5 and SERPINE/PAI-1 as two of the highly expressed genes. Which selection criteria did the authors use to in- or exclude differentially expressed genes in further analyses? Thus, in some instances, the rationale for the selection of the respective genes could be described more in detail in the manuscript.
- 2.) As far as I can see, neutrophil viability was assessed by measuring size and granularity (FSC, SSC) using flow cytometry (Figures 4 b,c). Did the authors include additional read-outs and methods to determine neutrophil viability (such as viability dyes)?
- 3.) In the part on the impact of the MUC1-ST macrophages on neutrophil functions (Figure 4 and Discussion), the authors mainly focus on pro-tumor neutrophil functions. Indeed, numerous recent studies show that neutrophils play major roles in linking inflammation and cancer and contribute to tumor progression and metastasis. However, neutrophils can also exhibit anti-tumor functions such as cytotoxicity or the activation of adaptive immune responses. In this regard, data shown in Figure 4 should be interpreted with care. The authors should discuss the ambivalent role of neutrophils in cancer more in detail.

Reviewer #1 (Remarks to the Author):

Dear Authors, Thank you for your thorough reply and improvements to the manuscript. I congratulate you on a very nice paper. I have no further comments.

Reviewer #3 (Remarks to the Author):

My (minor) concerns have been convincingly addressed; the manuscript can now be accepted for publication.

Reply: We thank the reviewers' for their very nice comments.